# Lycorine Carbamate Derivatives for Reversing P-glycoprotein-Mediated Multidrug Resistance in Human Colon Adenocarcinoma Cells

**DOI:** 10.3390/ijms24032061

**Published:** 2023-01-20

**Authors:** Shirley A. R. Sancha, Nikoletta Szemerédi, Gabriella Spengler, Maria-José U. Ferreira

**Affiliations:** 1Research Institute for Medicines (iMed.ULisboa), Faculty of Pharmacy, Universidade de Lisboa, Av. Prof. Gama Pinto, 1649-003 Lisbon, Portugal; 2Department of Medical Microbiology, Albert Szent-Györgyi Health Center, Faculty of Medicine, University of Szeged, Semmelweis utca 6, 6725 Szeged, Hungary

**Keywords:** *Pancratium maritimum*, Amaryllidaceae-type alkaloid, lycorine, carbamates, multidrug resistance, P-glycoprotein, ABCB1

## Abstract

Multidrug resistance (MDR) is a major challenge in cancer chemotherapy. Aiming at generating a small library of anticancer compounds for overcoming MDR, lycorine (**1**), a major Amaryllidaceae alkaloid isolated from *Pancratium maritimum*, was derivatized. Thirty-one new compounds (**2**–**32**) were obtained by chemical transformation of the hydroxyl groups of lycorine into mono- and di-carbamates. Compounds **1**–**32** were evaluated as MDR reversers, through the rhodamine-123 accumulation assay by flow cytometry and chemosensitivity assays, in resistant human colon adenocarcinoma cancer cells (Colo 320), overexpressing P-glycoprotein (P-gp, ABCB1). Significant inhibition of P-gp efflux activity was observed for the di-carbamate derivatives, mainly those containing aromatic substituents, at non-cytotoxic concentrations. Compound **5**, bearing a benzyl substituent, and compounds **9** and **25,** with phenethyl moieties, were among the most active, exhibiting strong inhibition at 2 µM, being more active than verapamil at 10-fold higher concentration. In drug combination assays, most compounds were able to synergize doxorubicin. Moreover, some derivatives showed a selective antiproliferative effect toward resistant cells, having a collateral sensitivity effect. In the ATPase assay, selected compounds (**2**, **5**, **9**, **19**, **25**, and **26**) were shown to behave as inhibitors.

## 1. Introduction

Cancer is a leading cause of mortality and morbidity worldwide. Despite all efforts made in the prevention, early detection, and treatment of cancer, the number of cases is expected to increase. According to the World Health Organization (WHO), by 2040, it is likely to increase to 29.4 million cases globally [1,2]. Multidrug resistance (MDR) to anticancer drugs is the major hurdle to successful cancer treatment not only with classical chemotherapy, but also with new targeted therapies, accounting for over 90% of deaths in patients with advanced cancer [3]. MDR can be inherent or acquired and can result from multiple mechanisms, such as decreased drug uptake, altered cell cycle checkpoints and cell cycle arrest, altered drug target, or increased drug efflux by drug transporters. However, the most significant mechanism results from the overexpression of ATP-binding cassette (ABC) drug transporter proteins. These transporters, which are generally overexpressed in cancer cells, use the energy of ATP to pump out chemotherapeutic drugs, consequently decreasing their intracellular concentration in the cells [4,5]. The main ABC transporter proteins associated with MDR are P-glycoprotein (P-gp or ABCB1), multidrug resistance-associated protein 1 (MRP1 or ABCC1), and breast cancer resistance protein (BCRP or ABCG2), P-gp being the most important. It is able to transport the main classes of anticancer drugs, such as anthracyclines, podophyllotoxins, and taxanes. Besides its role in anticancer drug resistance, P-glycoprotein is also crucial in the pharmacokinetics of drugs [4,6]. 

The development of P-gp inhibitors that can restore the cytotoxicity of the available anticancer drugs, when used in combination, has been considered among the most realistic approaches for overcoming P-gp-mediated MDR [7]. Several P-gp inhibitors have been developed and reached clinical trials. However, their clinical use has been hampered mainly due to the interference with pharmacokinetics of chemotherapeutic drugs, leading to toxicity [4]. Therefore, the development of effective P-gp inhibitors remains a great challenge. Due to these limitations, many researchers have been focused on natural products as a source of new and promising inhibitors [4].

Another strategy for overcoming MDR, known as collateral sensitivity (CS), focuses on searching for compounds that selectively exert more pronounced cytotoxicity against MDR cells than in the parental non-resistant cells from which they are derived. In this way, on MDR phenotypes, resistance to one cytotoxic agent can confer greater sensitivity to an alternate agent. The mechanisms by which some compounds exert a selective activity against MDR cells are still not clarified, although some CS agents have been recognized to have specific properties, such as the capability to: increase the production of reactive oxygen species (ROS); efflux endogenous substrates crucial for cell survival; increase sensitivity to changes in energy utilization of cells overexpressing ABC transporters; or induce plasma membrane perturbation [8,9]. 

Natural products have long been playing a crucial role in drug discovery and development. Around 60–70% of drugs on the market are of natural origin or their derivatives. In the case of anticancer drugs, they are mostly plant-derived compounds [4,10].

Plants from the Amaryllidaceae family, such as the *Pancratium* genus, have been proven to contain a range of unique biologically active compounds, the Amaryllidaceae-type alkaloids [11,12]. These compounds are exclusive to the family and are responsible for the pharmacological effects of these plants, including their strong anticancer properties [13]. They are also known for their activity as acetylcholinesterase inhibitors, such as the Amaryllidaceae alkaloid galantamine, which is clinically used for the treatment of Alzheimer’s disease [14]. 

In our search for new anticancer compounds for targeting multidrug resistance in cancer, our group has found several plant-derived compounds able to inhibit ABC transporters, e.g., [15,16,17,18,19,20,21,22]. Recently, we have identified a derivative of a monoterpene indole alkaloid, isolated from the African medicinal plant *Tabernaemontana elegans* [23], as an effective inhibitor of homologous DNA repair in triple-negative breast cancer and advanced ovarian cancer cells, acting by disrupting the breast cancer susceptibility protein (BRCA1) interaction with its binding partner BRCA1-associated ring domain protein (BARD1) [24]. 

Previously, we have isolated several Amaryllidaceae-type alkaloids from *Pancratium maritimum.* Some of them showed antiproliferative effect on triple-negative breast cancer cells. The homolycorine-type alkaloid 2α-10bα-dihydroxy-9-*O*-demethylhomolycorine, which was further studied, was able to induce apoptosis and arrest the cell cycle. Moreover, it exhibited synergistic effects with the chemotherapeutic drug etoposide [25]. 

In this study, aiming at generating new MDR-modifying compounds, thirty-one new derivatives (**2**–**31**) of lycorine (**1**), isolated in large amounts from *P. maritimum* [25], were prepared. Compounds **1**–**32** were evaluated for their activity as P-gp inhibitors in human adenocarcinoma cells by the rhodamine-123 accumulation assay. Moreover, the antiproliferative activity and the in vitro interaction between the compounds and the anticancer drug doxorubicin were addressed. The type of interaction between selected compounds and P-gp was also assessed through the ATPase assay.

## 2. Results

### 2.1. Chemistry

Aiming at preparing a small library of Amaryllidaceae-type alkaloids, lycorine (**1**), previously obtained in large amount from the alkaloid fraction of the methanol extract of the bulbs of *Pancratium maritimum* [25], was derivatized. Therefore, taking advantage of the hydroxyl groups at C-1 and C-2 positions of lycorine (**1**), several mono- and di-carbamates **2**–**32** (Figure 1) were prepared by reaction with carbonoylimidazole (CDI) and different aliphatic and aromatic amines.

Their structures were elucidated mainly by NMR data, including two-dimensional experiments (Appendix A). As expected, when comparing these data with the NMR spectra of the starting compound (**1**)**,** besides the extra proton and carbon signals corresponding to the carbamate moiety, the main differences were observed in the signals of ring C (Figure 1), which agreed with the effects expected for this chemical modification. In this way, in the ^1^H NMR spectra of the mono-carbamates, a significant paramagnetic effect was observed in the chemical shift of H-2, which appeared at 1.3 to 1.6 ppm downfield. Similarly, in the ^13^C NMR spectra, deshielding effects were observed in the chemical shift of C-2 (Δδ_C_ + 2.1 to 3.3 ppm, α-carbon), C-4, and C-10b (Δδ_C_ + 2.9 to 4.3, and + 0.7 to 1.7 ppm, respectively, γ-carbons), together with diamagnetic effects at C-1 and C-3 (Δδ_C_ − 0.7 to 1.4, and −2.3 to 4.0 ppm, respectively, β-carbons). Regarding the di-carbamate derivatives, besides the difference in the chemical shift of H-2, the signal of H-1 was also shifted downfield (Δδ_H_ + 0.7 to 1.4 ppm). In the same way, in ^13^C NMR spectra, a paramagnetic effect at the γ-carbons (C-4 and C-4a) and diamagnetic effect at β-carbon (C-3) were observed.

### 2.2. Biological Activity

#### 2.2.1. Antiproliferative Activity and Collateral Sensitivity Effect

The antiproliferative activity of lycorine (**1**) and the carbamate derivatives (**2**–**32**) was evaluated through the thiazolyl blue tetrazolium bromide (MTT) assay in sensitive and resistant human colon adenocarcinoma cancer cells (Colo205, sensitive; Colo320, resistant). In addition, the antiproliferative activity was also evaluated in normal lung fibroblast cells (MRC-5). The results, obtained as the concentration of the compound that produced half of the inhibition (IC_50_), are shown in Table 1. As it can be observed, the lowest IC_50_ values were observed in lycorine (**1**) for both sensitive and resistant cells (IC_50_ values of 1.06 and 0.93 µM, for sensitive Colo205 and resistant Colo320 adenocarcinoma cells, respectively). Conversely, in the sensitive cancer cell line (Colo205), the derivatives were inactive or barely active, displaying IC_50_ values ranging from 19.3 to 96.2 µM. However, when analyzing the results against the resistant subline (Colo320), some compounds (**5**, **9**, **16**, **19**, **25**, **26**) were found to be more active against the resistant cells than the sensitive cells, with compounds **9**, **19**, and **25** displaying simultaneously strong antiproliferative activity (**9**, IC_50_ = 10.39; **19**, IC_50_ = 7.64; **25**, IC_50_ = 3.14). 

The IC_50_ values obtained in normal lung fibroblast cells (MRC-5) were compared with those obtained for sensitive and resistant colon cancer cells by determining the selectivity index (SI) values. As it can be observed in Table 1, in resistant cancer cells, the highest selective index was found for compound **19** (SI_C/B_ = 5.66). 

To evaluate the potential collateral sensitivity effect of compounds, the relative resistance (RR) values were calculated as the ratio between the IC_50_ of a compound against the resistant cells and the IC_50_ against the corresponding sensitive cells. Compounds with RR < 1 show selectivity against resistant cells, whereas RR ≤ 0.5 means that the CS effect occurs [8]. As it can be observed in Table 1, when analyzing the RR values, compounds **5**, **9**, **13**, **16**, **25**, and **26** showed selectivity to the resistant cells (RR < 1), whereas compounds **5** (RR < 0.3), **9** (RR < 0.10), **16** (RR = 0.48), and **25** (RR = 0.065) showed CS effect. 

#### 2.2.2. Inhibition of P-glycoprotein Efflux Activity 

Compounds **1**–**32** were investigated for their ability to inhibit the P-gp transporter by measuring the intracellular accumulation of its fluorescent substrate rhodamine-123 in human adenocarcinoma cancer cells, both sensitive (Colo205) and resistant cells (Colo320), using the standard rhodamine-123 functional assay. The compounds were tested at 2 and 20 µM, and compound **1** was also tested at 0.2 µM. Verapamil, a standard P-gp inhibitor, was used as a positive control (20 µM). Fluorescence activity ratio (FAR) values were calculated by determining the quotient between the intracellular accumulation of rhodamine-123 in resistant and sensitive cancer cells. Compounds with FAR values higher than 1 are able to decrease rhodamine-123 efflux in resistant cells and are considered P-gp inhibitors. The results are summarized in Table 2. As shown, the inhibitory activity of the derivatives **2**–**32** was dose-dependent, with a significant increase in most of the compounds when compared with the parental compound **1**, which was barely active.

Excepting derivatives **15, 29,** and **30**, at 20 µM most of the compounds were found to have P-gp inhibitory activity with FAR values ranging from 2.69 to 21.78. At 2 µM, compounds **5**, **9**, and **25** (FAR = 9.93, 12.77, and 15.84, respectively) were the most active, having FAR values higher (up 1.2 to 1.8-fold) than verapamil (FAR = 7.78 at 20 µM) at 10-fold higher concentration. Strong inhibitory activity (FAR values > 10) was also observed in compounds **5**, **7**, **9**, **20**, **22**, **23**, **25**, and **27,** at 20 µM**,** exhibiting FAR values significantly higher (up 1.2 to 2.8-fold) than verapamil at the same concentration.

#### 2.2.3. Chemosensitivity Assays

The interaction of lycorine (**1**) and some selected derivatives (**2**, **5**, **9**, **13**, **16**, **19**, **25**, and **26**) in combination with the chemotherapeutic drug doxorubicin was performed by the checkerboard assay, a widely used technique for assessing drug interactions, in resistant human Colo320 colon adenocarcinoma cells. The type of drug–drug interaction was evaluated by using the Chou and Talalay method for drug combination and expressed using the combination index (CI) values, and therefore evaluated as synergism (CI < 1), additive (CI = 1), or antagonist (CI > 1) [27,28]. 

As shown in Figure 1, all the tested compounds interacted synergistically with doxorubicin (CI < 1), with exception of the parental compound **1**, which behaved as a doxorubicin antagonist (CI = 1.3). The derivative **5** showed the strongest synergistic effect (CI = 0.13). 

#### 2.2.4. P-gp ATPase Activity

To evaluate the type of interaction between selected compounds and P-gp, namely to determine if they interacted as stimulators or inhibitors, the ATPase activity assay (P-gp.Glo^TM^) [29] was performed, using human P-gp membranes. This assay is based on the ATP consumption necessary for the efflux, obtained from ATP hydrolysis by light-generating firefly luciferase reaction. The P-gp basal activity (100% inhibition observed) is given by the difference between the luminescence of the samples treated with sodium orthovanadate (vanadate, Na_3_VO_4_) and that of those untreated. Vanadate is a selective and strong P-gp inhibitor. Therefore, the compounds can be classified as inhibitors or stimulators/substrates by comparing P-gp ATPase activity with the basal activity; stimulators give results higher than 100%, while inhibitors result below 100% of the basal activity. Verapamil (0.5 mM), a known P-gp substrate that stimulates ATPase activity, was used as a control [29]. The results are presented in Figure 2 as the difference between the luciferase luminescence observed when treated with the tested compounds and with vanadate and compared with the basal activity (100%). 

As shown in Figure 2, at the tested concentration, all the selected compounds (**2, 5, 9, 19, 25, 26**) decreased the basal activity, consequently inhibiting the P-gp ATPase activity.

## 3. Discussion

The development of inhibitors of ABC transporters, namely P-gp inhibitors, has been considered a good approach for tackling MDR, one of the major challenges of successful chemotherapy in cancer treatment. Therefore, aiming at finding effective P-gp inhibitors, the Amaryllidaceae-type alkaloid lycorine (**1**) was derivatized by introducing the carbamate group. Considered a structural analog of the amid function, which has limitations in drug development due to pharmacokinetic issues, the carbamate moiety is widely used in drug design and discovery, appearing in the structure of many therapeutic agents, owing to favorable chemical properties namely its stability, ability to increase permeability across cellular membranes, and its hydrogen bonding potential [30].

When analyzing the results obtained in the rhodamine accumulation assay (Table 2), the most active compounds were di-carbamate derivatives (**5**, **7**, **9**, **20**, **22**, **23**, **25**, **27**, and **32**). The highest FAR values were exhibited by those sharing phenethyl (**9**, **23**, **25**) or benzyl moieties (**5**, **20**, **22**, **27**). Compound **25**, bearing a phenethyl motif with an electron-withdrawing substituent at *para* position, was the strongest inhibitor (FAR = 15.84 and 21.78, at 2 and 20 µM, respectively), being at 2 µM 2-fold more active than verapamil at 20 µM. When comparing the FAR values of compounds **27** (FAR = 4.70 and 11.41, at 2 and 20 µM, respectively) and **32** (FAR = 2.52 and 9.48, at 2 and 20 µM, respectively), containing benzyl moieties, with the corresponding derivatives **9** (FAR = 12.77 and 17.92, at 2 and 20 µM, respectively) and **23** (FAR = 6.93 and 14.84, at 2 and 20 µM, respectively), sharing phenethyl substituents, the significant increase in the activity of the last compounds might be related to the higher flexibility of the phenethyl group, which also increase the lipophilicity. Unexpectedly, compound **4** (FAR = 1.75 and 3.20, at 2 and 20 µM, respectively), bearing the electron-donating *para*-methoxy group in the benzene ring able to increase its hydrogen-bonding potential, was much less active than compound **9** (FAR = 12.77 and 17.92, at 2 and 20 µM, respectively), without substituents, whereas the introduction of chloride in compound **25** (FAR = 15.84 and 21.78, at 2 and 20 µM, respectively), was beneficial for the activity. Dicarbamates with aliphatic moieties, such as compounds **14**, **16**, and **29**, showed no significant activity, thus emphasizing the importance of the aromatic moieties for the activity. The importance of this structural feature was supported by the results obtained for dicarbamates, where a second benzene moiety may establish additional π–π stacking interactions between the aromatic motifs and the amino acid residues in the P-gp binding site [31,32]. 

Since P-gp efflux activity has been described as dependent on the energy from adenosine triphosphate (ATP) hydrolysis, the interaction of some representative compounds (**2**, **5**, **9**, **19**, **25**, and **26**) with this ABC transporter was further investigated, using human P-gp membranes, by analyzing the ATPase activity [29]. All the selected derivatives inhibited the P-gp ATPase activity. Consequently, they may act directly by blocking the ATP hydrolysis, binding to the P-gp ATP binding site (non-competitive inhibition), or by binding to a P-gp allosteric residue, reducing its efflux activity and consequently inhibiting the ATPase activity.

The anti-MDR activity of selected compounds was supported by their ability to synergize doxorubicin (CI < 1) in resistant cells. It should be noted that compounds **13** and **16**, which showed weak inhibitory activity, could also synergistically improve the cytotoxicity of doxorubicin, suggesting that other mechanisms might be involved.

Furthermore, compounds **5**, **9**, **16**, **19**, and **25** revealed to be selective against resistant adenocarcinoma cells (Colo320), thus having collateral sensitivity effect (RR ≤ 0.5). Taking together the antiproliferative activity and the collateral sensitivity effect, the best results were found for the dicarbamate **25**, showing both the strongest antiproliferative effect and the highest selectivity against resistant cells (IC_50_ = 3.14; RR = 0.065)**.** Interestingly, compound **25** also showed the highest P-gp inhibitory activity, thus having a dual role in reversing P-gp-mediated resistance. Similarly, compounds **9** (IC_50_ = 10.39; RR < 0.1) and **19** (IC_50_ = 7.64; RR = 0.28) also showed simultaneously significant antiproliferative effect, compound **9** also being one of strongest P-gp inhibitors.

### Prediction of Physicochemical and Pharmacokinetic Properties 

To find correlations between FAR values and some physicochemical properties of compounds **1**–**32**, the molecular descriptors molecular weight (MW), logarithm of the octanol/water partition coefficient (log *P*), topological polar surface area (TPSA), and H-bond acceptors (HBA) and donors (HBD) were calculated (Appendix A). As it can be observed in Appendix A, the three most active compounds share TPSA values (98.3 Å^2^), and number of H-bond acceptors (7) and donors (2). Taking into account that log *P* values higher than 2.92 are important for the P-gp inhibitory activity [32], their high lipophilicity (**5**, log *P* = 5.23; **9**, log *P* = 4.70; **25**, log *P* = 6.0) may be also associated with their activity as P-gp inhibitors. This assumption is corroborated by the lower activity exhibited by the corresponding mono-carbamates (**3**, **6**, **10**, **15**, and **30**) with log *P* values ranging from 1.22 to 2.99. However, poor linear correlation was found between log *P* and FAR values. 

Several factors can influence the therapeutic efficacy of a drug, including pharmacokinetics, which is characterized mainly by its absorption, distribution, metabolism, and excretion (ADME) properties. Thus, along with pharmacodynamics, it is also important to study the pharmacokinetics of a compound in early stages. Therefore, pharmacokinetic parameters were predicted for the reference inhibitor, verapamil, and the derivatives, using the pkCSM descriptor’s algorithm protocol (Appendix A) [33]. 

All the compounds were predicted to be moderately soluble, expressed as Log *S*, (solubility of a compound in water at 25°; −4.9 < log *S* < −2.5), and most of them were predicted to have good intestinal absorption (94.4–100%), better than verapamil (94.2%). Furthermore, most of the derivatives were indicated to have Caco-2 permeability with values greater than 0.9, which is the minimum value established for high permeability in a predictive model [33]. The fractional unbound (fu) estimated showed that most of the relevant derivatives (**5**, **9**, **20**, **22**, **25**, and **32**) and verapamil had a fractional unbound lower than the minimum recommended value of 0.01 fu [34], whereas the remaining derivatives showed higher values, between 0.04 and 0.42 fu. None of the derivatives was able to penetrate the central nervous system (CNS permeability < −3). Furthermore, most of the derivatives were predicted to inhibit the cytochrome P3A4.

## 4. Materials and Methods

### 4.1. General Experimental Procedures 

The NMR spectra were recorded on a Bruker 300 Ultra-Shield instrument (^1^H 300 MHz, ^13^C 75 MHz), using CDCl_3_, CD_3_OD, or DMSO-d_6_ as solvents. Chemical shifts in ^1^H and ^13^C NMR spectra are expressed in parts per million (ppm) on the δ scale using the solvent signal as an internal standard. NMR spectra were assigned using appropriate DEPT, COSY, HSQC, and HMBC sequences. Low-resolution mass spectrometry was performed in a Triple Quadrupole mass spectrometer from Waters AcquirityTM (Waters^®^, Ireland). The solvents used were dried according to published methods and distilled before use. Other reagents obtained from commercial suppliers were used without further purification. Column chromatography (CC) was performed on silica gel (Merck 9385, Darmstadt, Germany). Analytical thin-layer chromatography (TLC) was performed on precoated Merck silica gel 60 F_254_ plates with visualization under UV light (λ 254 and 366 nm) by spraying either with Dragendorff’s reagent or a solution of H_2_SO_4_ and MeOH (1:1), followed by heating.

### 4.2. Tested Compounds 

Lycorine (**1**) and the carbamates (**2**–**32**), whose preparation is described below, were evaluated for their MDR reversal activity.

#### 4.2.1. General Preparation of Lycorine Carbamates **2**–**32**

1,1′-carbonyldiimidazole (CDI, 3 equiv.) was added to a solution of lycorine (**1**, 50–78 mg, 1 equiv.) in dimethylformamide (DMF, 3 mL) and stirred for 30 min under nitrogen atmosphere at 50 °C. Then, the corresponding amine (3 equiv.) was added, and the reaction mixture stirred for 1 to 2 h. After the complete reaction, the solvent was removed, and the crude residue was purified by column chromatography.

##### Lycorine 2-yl (4-chlorobenzyl)carbamate (**2**)

Obtained from the reaction between lycorine (52 mg, 0.189 mmol, 1 Equiv.), CDI (88 mg, 0.54 mmol, 3 Equiv.), and 4-chlorobenzylamine (66.3 µL, 0.54 mmol, 3 Equiv.). The residue was purified by flash chromatography (silica gel CH_2_Cl_2_/MeOH, 100:0 to 95:5) to afford 47 mg of compound **2** (0.103 mmol, yield 57%) as an amorphous powder. ^1^H NMR (300 MHz, CDCl_3_/CD_3_OD) 7.31 (2H, *bd*, *J* = 8.3 Hz, H-6′, H-8′), 7.25 (2H, *bd*, *J* = 8.4 Hz, H-5′, H-9′), 6.83 (1H, *s*, H-10), 6.61 (1H, *s*, H-7), 5.91 (2H, *s*, OCH_2_O), 5.53 (1H, *bs*, H-3), 5.19 (1H, *bs*, H-2), 4.52 (1H, *bs*, H-1), 4.28 (2H, *bs*, H-3′), 4.11 (1H, *d*, *J* = 14.1 Hz, H-6β), 3.54 (1H, *d*, *J* = 14.3 Hz, H-6α), 3.40–3.29 (1H, *m*, H-12β), 2.91 (1H, *d*, *J* = 10.7 Hz, H-4a), 2.67 * (1H, *bd*, *J* = 10.1 Hz, H-10b), 2.66–2.52 * (2H, *m*, H-11), 2.45 (1H, *q*, *J* = 8.9 Hz, H-12α) ppm, * overlapped signals.^13^C NMR (75 MHz, CDCl_3_/CD_3_OD) δ 156.9 (C-1′), 146.7 (C-8), 146.3 (C-9), 144.7 (C-4), 137.7 (C-4′), 132.6 (C-7′), 128.8 (C-10a), 128.5 (C-5′, C-9′), 128.2 (C-6′, C-8′), 127.7 (C-6a), 114.6 (C-3), 106.9 (C-7), 104.6 (C-10), 100.9 (-OCH_2_O), 73.8 (C-2), 68.1 (C-1), 60.8 (C-4a), 56.5 (C-6), 53.3 (C-12), 43.6 (C-3′), 40.8 (C-10b), 28.1 (C-11) ppm. ESI-MS m/z 455 [M + H] ^+^.

##### Lycorine 2-yl morpholine-4-carboxylate (**3**)

Obtained from the reaction between lycorine (78 mg, 0.27 mmol, 1 equiv.), CDI (132 mg, 0.81 mmol, 3 equiv.), and morpholine (70 µL, 0.81 mmol, 3 equiv.). The residue was purified by column chromatography (silica gel CH_2_Cl_2_/MeOH, 100:0 to 95:5) to afford 41 mg of compound **3** (0.10 mmol, yield 37%) as an amorphous powder. ^1^H NMR (300 MHz, CDCl_3_) δ 6.83 (1H, *s*, H-10), 6.58 (1H, *s*, H-7), 5.91 (2H, *dd*, *J* = 4.2, 1.4 Hz, OCH_2_O), 5.47 (1H, *bs*, H-3), 5.26 (1H, *bs*, H-2), 4.55 (1H, *bs*, H-1), 4.14 (1H, *d*, *J* = 14.1 Hz, H-6β), 3.71–3.58 * (4H, *m*, H-4′, H-6′), 3.53 * (1H, *d*, *J* = 14.0 Hz, H-6α), 3.49–3.79 * (4H, *m*, H-3′, H-7′), 3.35 (1H, *dd*, *J* = 8.8, 4.7 Hz, H-12β), 2.82 (1H, *d*, *J* = 10.5 Hz, H-4a), 2.67 * (1H, *m*, H-10b), 2.64 * (2H, *m*, H-11), 2.38 (1H, *q*, *J* = 8.8 Hz, H-12α) ppm, * overlapped signals. ^13^C NMR (75 MHz, CDCl_3_) δ 154.9 (C-1′), 146.5 (C-8), 146.2 (C-9), 145.8 (C-4), 129.8 (C-10a), 127.4 (C-6a), 114.0 (C-3), 107.5 (C-7), 104.8 (C-10), 100.9 (-OCH_2_O), 74.8 (C-2), 69.0 (C-1), 66.4 (C-4′, C-6′), 60.7 (C-4a), 56.9 (C-6), 53.7 (C-12), 44.3 (C-3′, C-7′), 40.8 (C-10b), 28.6 (C-11) ppm. ESI-MS m/z 401 [M + H]^+^.

##### Lycorine 1,2-diyl bis [(4-methoxyphenethyl)carbamate] (**4**)

Obtained from the reaction between lycorine (50 mg, 0.17 mmol, 1 equiv.), CDI (84 mg, 0.52 mmol, 3 equiv.), and 4-methoxy phenethylamine (76 µL, 0.52 mmol, 3 equiv.). The residue was purified by column chromatography (silica gel CH_2_Cl_2_/MeOH, 100:0 to 95:5) to afford 12 mg of compound **4** (0.019 mmol, yield 10%) as an amorphous powder. ^1^H NMR (300 MHz, CDCl_3_) δ 7.16–7.05 * (3H, *m*, H-6′, H-10′, H-6′′), 6.95 (1H, *bd*, *J* = 8.2 Hz, H-10′′), 6.90–6.81 * (2H, *m*, H-7′, H-7′′, H-10), 6.84 * (1H, *s*, H-10), 6.77 * (2H, *m*, H-9′, H-9′′), 6.5 (1H, *s*, H-7), 5.89 (2H, *dd*, *J* = 10.1, 1.4 Hz, OCH_2_O), 5.64 (1H, *bs*, H-1), 5.56 (1H, *bs*, H-2), 5.22 (1H, *bs*, H-3), 4.73 (1H, *bs*, -NH), 4.55 (1H, *t*, *J* = 6.1 Hz, -NH), 4.13 (1H, *d*, *J* = 14.1 Hz, H-6), 3.78 * (4H, *s*, OCH3, H-6α) 3.77 * (3H, *s*, OCH_3_), 3.53 (1H, *bd*, *J* = 7.0 Hz, H-12β), 3.48–3.27 (4H, *m*, H-3′, H-3′′), 2.81–2.57 (8H, *m*, H-10b, H-4a, H-11, H-4′, H-4′′), 2.38 (1H, *q*, *J* = 6.1 Hz, H-12α) ppm.^13^C NMR (75 MHz, CDCl_3_) δ 158.3(C-1′′), 158.2 (C-1′), 155.1 (C-8′, C-8′′), 146.5 (C-8, C-9), 131.1 (C-5′′, C-5′), 130.6 (C-10a), 129.7 (C-6′, C-10′), 127.1 (C-6a), 114.1 (C-7′. C-7′′), 114.0 (C-9′. C-9′’), 113.9 (C-3), 107.7 (C-10), 105.3 (C-10), 101.1 (OCH_2_O), 70.8 (C-2), 69.2 (C-1), 56.9 (C-6), 55.2 (OCH_3_), 53.8 (C-12), 42.4 (C-3′, C-3′′), 41.8 (C-10b), 35.2 (C-4′, C-4′′), 28.8 (C-11) ppm. ESI-MS m/z 582 [M + H]^+^.

##### Lycorine 1,2-diyl bis[(4-methylbenzyl)carbamate] (**5**) and lycorine 2-yl (4-methylbenzyl)carbamate (**6**)

Obtained from the reaction between lycorine (50 mg, 0.17 mmol, 1 equiv.), CDI (84 mg, 0.52 mmol, 3 equiv.), and 4-methyl benzylamine (66.2 µL, 0.52 mmol, 3 equiv.). The residue was purified by column chromatography (silica gel CH_2_Cl_2_/MeOH, 100:0 to 95:5) to afford 31 mg of compound **5** (0.052 mmol, yield 29.9%) and 42 mg of compound **6** (0.096 mmol, yield 55.5%) as amorphous powders.

Lycorine 1,2-diyl bis[(4-methylbenzyl)carbamate] (**5**): ^1^H NMR (300 MHz, CDCl_3_) δ 7.22–7.03 (8H, *m*, H-Ar), 6.86 (1H, *s*, H-10), 6.58 (1H, *s*, H-7), 5.91 (2H, *s*, OCH_2_O), 5.71 (1H, *bs*, H-1), 5.59 (1H, *bs*, H-3), 5.29 (1H, *bs*, H-2), 5.10 (1H, *bs*, -NH), 4.85 (1H, *t*, *J* = 5.9 Hz, -NH), 4.33 (2H, *dd*, *J* = 8.7, 5.1 Hz, H-3′), 4.26 (2H, *bd*, *J* = 8.2 Hz, H3′′), 4.11 (1H, *d*, *J* = 14.1 Hz, H-6β), 3.47 (1H, *d*, *J* = 14.1 Hz, H-6α), 3.32 (1H, *dt*, *J* = 9.2, 4.8 Hz, H-12β), 2.80 (1H, *d*, *J* = 10.5 Hz, H-4a), 2.70 (1H, *d*, *J* = 10.7 Hz, H-10b) 2.65 (2H, *bs*, H-11), 2.38 * (1H, *bd*, *J* = 8.7 Hz, H-12α), 2.33 * (3H, *s*, -CH_3_), 2.31 * (3H, *s*, -CH*_3_*) ppm, * overlapped signals. ^13^C NMR (75 MHz, CDCl_3_) δ 155.3 (C-1′, C-1′′), 146.4 (C-9), 146.2 (C-8), 145.4 (C-4), 137.1 (C-7′, C-7′′), 135.3 (C-4′), 135.2 (C-4′′), 129.4 (C-10a), 129.3 (C-6′, C-6′′), 129.2 (C-8′, C-8′′), 127.6 (C-5′, C-9′), 127.5 (C-5′′, C-9′′), 127.0 (C-6a), 114.7 (C-3), 107.2 (C-7), 105.4 (C-10), 100.9 (OCH_2_O), 71.5 (C-2), 70.2 (C-1), 61.3 (C-4a), 56.9 (C-6), 53.6 (C-12), 44.8 (C-3′, C-3′′), 40.5 (C-10b), 28.6 (C-11), 21.0 (CH_3_) ppm. ESI-MS m/z (rel.int) 582 [M + H]^+^.

Lycorine 2-yl (4-methylbenzyl)carbamate (**6**): ^1^H NMR (300 MHz, CDCl_3_) δ 7.15 (4H, *q*, *J* = 7.8 Hz, H-Ar), 6.79 (1H, *s*, H-10), 6.58 (1H, *s*, H-7), 5.90 (2H, *dd*, *J* = 8.8, 1.5 Hz, -OCH_2_O), 5.49 (1H, *s*, H-3), 5.30 (1H, *d*, *J* = 6.1 Hz, -NH), 5.23 (1H, *bs*, H-2), 4.52 (1H, *bs*, H-1), 4.31 (2H, *bd*, *J* = 5.9 Hz, H-3′), 4.08 (1H, *d*, *J* = 14.0 Hz, H-6β), 3.56 (1H, *d*, *J* = 14.0 Hz, H-6α), 3.33 (1H, *dt*, *J* = 9.4, 4.9 Hz, H-12β), 2.89 (1H, *d*, *J* = 10.7 Hz, H-4a), 2.71–2.58 * (3H, *m*, H-10b, H-11), 2.46 (1H, *q*, *J* = 8.6 Hz, H-12α), 2.33 (3H, *s*, CH_3_) ppm, * overlapped signals. ^13C^ NMR (75 MHz, CDCl_3_) δ 156.0 (C-1′), 146.6 (C-8), 146.3 (C-9), 145.0 (C-4), 137.2 (C-7′), 135.2 (C-5′), 129.3 (C-6′, C-8′, C-10a), 127.6 (C-6a), 127.5 (C-5′, C-9′), 114.5 (C-3), 107.6 (C-7), 104.9 (C-10), 100.9 (-OCH_2_O), 74.2 (C-2), 69.0 (C-1), 56.6 (C-6), 53.7 (C-12), 44.9 (C-3′), 41.2 (C-10b), 21.0 (CH_3_) ppm. ESI-MS m/z 435 [M + H]^+^.

##### Lycorine 1,2-diyl bis[(piperonyl)carbamate] (**7**) and lycorine 2-yl (piperonyl)carbamate) (**8**)

Obtained from the reaction between lycorine (50 mg, 0.17 mmol, 1 equiv.), CDI (84 mg, 0.52 mmol, 3 equiv.), and piperonylamine (65.2 µL, 0.52 mmol, 3 equiv.). The residue was purified by column chromatography (silica gel CH_2_Cl_2_/MeOH, 100:0 to 96:4) to afford 14 mg of compound **7** (0.18 mmol, yield 10.7%) and 40 mg of compound **8** (0.86 mmol, 49.4%) as yellow amorphous and amorphous powders.

Lycorine 1,2-diyl bis[(piperonyl)carbamate] (**7**): ^1^H NMR (300 MHz, CDCl_3_) δ 6.85 (1H, *s*, H-5′′), 6.80 (1H, *s*, H-5′), 6.75 (2H, *bs*, H-8′, H-8′′), 6.70 (1H, *bd*, *J* = 8.8 Hz, H-9′′), 6.67 (1H, *s*, H-10), 6.62 (1H, *bd*, *J* = 7.7 Hz, H-9′), 6.54 (1H, *s*, H-7), 5.93 (2H, *s*, -OCH_2_O), 5.91 (4H, *bd*, *J* = 1.6 Hz, -OCH_2_O), 5.69 (1H, *bs*, H-1), 5.58 (1H, *bs*, H-3), 5.29 (1H, *bs*, H-2), 5.06 (1H, *bs*, -NH), 4.83 (1H, *bt*, *J* = 6.2 Hz, -NH), 4.31- 4.24 * (2H, *m*, H-3′), 4.21 * (2H, *t*, *J* = 7.0 Hz, H-3′′), 4.12 * (1H, *d*, *J* = 13.9 Hz, H-6β), 3.47 (1H, *d*, *J* = 14.2 Hz, H-6α), 3.33 (1H, *dd*, *J* = 9.1, 4.7 Hz, H-12β), 2.79 (1H, *d*, *J* = 10.5 Hz, H-4a), 2.70 (1H, *bd*, *J* = 10.9 Hz, H-10b), 2.60 (2H, *bs*, H-11), 2.42–2.30 (1H, *m*, H-12α) ppm, * overlapped signals.^13^C NMR (75 MHz, CDCl_3_) δ 155.3 (C-1′, C-1′′), 148.0 (C-6′, C-6′′), 147.1 (C-7′, C-7′′), 146.4 (C-8, C-9), 132.3 (C-4′, C-4′′), 129.4 (C-10a), 127.0 (C-6a), 121.0 (C-8′, C-8′′), 114.7 (C-3), 108.4 (C-5′, C-5′′), 108.3 (C-8′, C-8′′), 107.3 (C-7), 105.5 (C-10), 101.1 (-OCH_2_O), 101.1 (-OCH_2_O), 101.1 (-OCH_2_O), 71.7 (C-2), 70.4 (C-1), 61.4 (C4a), 57.1 (C-6), 53.8 (C-12), 45.2 (C-3′), 45.1 (C-3′′), 40.6 (C-10b), 28.5 (C-11) ppm. ESI-MS m/z 642 [M + H]^+^. 

Lycorine 2-yl (piperonyl)carbamate (**8**): ^1^H NMR (300 MHz, CDCl_3_) δ 6.77 * (2H, *bs*, H-10, H-5′), 6.73 (2H, *s*, H-8′, H-9′), 6.57 (1H, *s*, H-7), 5.93 (2H, *bs*, -OCH_2_O), 5.90 (2H, *dd*, *J* = 8.9, 1.5 Hz, -OCH_2_), 5.47 (1H, *bs*, H-3), 5.36 (1H, *t*, *J* = 6.0 Hz, -NH), 5.22 (1H, *bs*, H-2), 4.49 (1H, *s*, H-1), 4.24 (1H, *bd*, *J* = 5.9 Hz, H-3′), 4.10 (1H, *d*, *J* = 13.6 Hz, H-6β), 3.50 (1H, *d*, *J* = 13.8 Hz, H-6α), 3.32 (1H, *dt*, *J* = 9.1, 4.8 Hz, H-12α), 2.81 (1H, *d*, *J* = 10.6 Hz, H-4a), 2.63 (3H, *bd*, *J* = 9.7 Hz, H-10b, H-11), 2.36 (1H, *q*, *J* = 8.8 Hz, H-12α) ppm, * overlapped signals. ^13^C NMR (75 MHz, CDCl_3_) δ 156.2 (C-1′), 148.0 (C-8′), 147.1 (C-6′), 146.6 (C-8), 146.3 (C-9), 145.7 (C-4), 132.4 (C-4′), 129.8 (C-10a), 127.7 (C-6a), 120.9 (C-9′), 114.3 (C-3), 108.4 (C-8′), 108.3 (C-5′), 107.7 (C-7), 104.9 (C-10), 101.1 (-OCH_2_O), 101.0 (-OCH_2_O), 74.5 (C-2), 69.3 (C-1), 60.8 (C-4a), 57.0 (C-6), 53.8 (C-12), 45.0 (C-3′), 41.6 (C-10b), 28.8 (C-11) ppm. ESI-MS m/z 465 [M + H]^+^.

##### Lycorine 1,2-diyl bis(phenethylcarbamate) (**9**) and lycorine 2-yl phenethylcarbamate (**10**)

Obtained from the reaction between lycorine (50 mg, 0.17 mmol, 1 equiv.), CDI (84 mg, 0.52 mmol, 3 equiv.), and 2-phenethylamine (65.7 µL, 0.52 mmol, 3 equiv.). The residue was purified by column chromatography (silica gel CH_2_Cl_2_/MeOH, 100:0 to 96:4) to afford 33 mg of compound 9 (0.056 mmol, yield 32.6%) and 48 mg of compound 10 (0.11 mmol, 63.4%) as amorphous powders.

Lycorine 1,2-diyl bis(phenethylcarbamate) (**9**): ^1^H NMR (300 MHz, CDCl_3_) δ 7.34–7.15 (8H, *m*, H-Ar), 7.04 (2H, *bd*, *J* = 7.0 Hz, H-Ar), 6.87 (1H, *s*, H-10), 6.56 (1H, *s*, H-7), 5.90 (2H, *bd*, *J* = 5.8 Hz, -OCH_2_O), 5.65 (1H, *bs*, H-1), 5.56 (1H, *bs*, H-2), 5.23 (1H, *bs*, H-3), 4.78 (1H, *bs*, NH), 4.58 (1H, *t*, *J* = 6.2 Hz, NH), 4.13 (1H, *d*, *J* = 14.1 Hz, H-6β), 3.49 * (1H, *d*, *J* = 14.4 Hz, H-6α), 3.47–3.36 * (4H, *m*, H-3′, H-3′′), 3.36–3.27 * (1H, *m*, H-12β), 2.87 -2.79 * (3H, *m*, H-4a, H-4b’, H-4b’’), 2.78–2.68 * (3H, *m*, H-4ª, H-4′, H-4′′), 2.65–2.58 * (3H, *m*, H-11, H-4′, H-4′′), 2.36 (1H, *q*, *J* = 8.7 Hz, H-12α) ppm, * overlapped signals.^13^C NMR (75 MHz, CDCl_3_) δ 155.4 (C-1′), 155.3 (C-1′′), 146.6 (C-8), 146.4 (C-9), 145.4 (C-3), 139.3 (C-5′), 138.9 (C-5′′), 129.5 (C-10a), 129.0 (C-6′, C-10′), 128.9 (C-8′, C-8′′), 128.8 (C-9′), 128.7 (C-9′′), 128.6 (C-6′′, C-10′′), 127.1 (C-6a), 126.6 (C-7′), 126.5 (C-7′′), 114.8 (C-3), 107.3 (C-7), 105.6 (C-10), 101.0 (-OCH_2_O), 71.5 (C-2), 70.1 (C-1), 61.4 (C-4a), 57.0 (C-6), 53.8 (C-12), 42.4 (C-3′), 41.7 (C-3′′), 36.5 (C-4′, C-4′′), 29.0 (C-11) ppm. ESI-MS m/z 582 [M + H]^+^. 

Lycorine 2-yl phenethyl) carbamate (**10**): ^1^H NMR (300 MHz, CDCl_3_) δ 7.36 -7.14 * (5H, *m*, H-Ar), 6.78 (1H, *s*, H-10), 6.58 (1H, *s*, H-7), 5.88 (2H, *bd*, *J* = 7.0 Hz, -OCH_2_O), 5.45 (1H, *bs*, H-1), 5.19 (1H, *bs*, H-3), 5.04 (1H, *bs*, -NH), 4.46 (1H, *bs*, H-1), 4.12 (1H, *d*, *J* = 14.2 Hz, H-6β), 3.52 * (1H, *d*, *J* = 14.3 Hz, H-6α), 3.48–3.39 * (1H, *m*, H-3′), 3.32 (1H, *dd*, *J* = 9.1, 4.7 Hz, H-12β), 2.82 * (3H, *bd*, *J* = 7.9 Hz, H-4′, H-4a), 2.61 * (3H, *bd*, *J* = 9.4 Hz, H-10b, H-11), 2.37 (1H, *q*, *J* = 9.0 Hz, H-12α) ppm, * signals overlapped. ^13^C NMR (75 MHz, CDCl_3_) δ 156.0 (C-1′), 146.5 (C-8), 146.2 (C-9), 145.5 (C-4), 138.5 (C-5′), 129.7 (C-10a), 129.5 (C-7′), 128.7 (C-6′), 128.6 (C-10′), 128.0 (C-8′), 127.6 (C-6a), 126.5 (C-6′), 114.2 (C-3), 107.5 (C-7), 104.8 (C-10), 100.9 (-OCH_2_O), 74.2 (C-2), 69.1 (C-1), 60.7 (C-4a), 56.9 (C-6), 53.7 (C-12), 42.2 (C-3′), 36.0 (C-4′), 28.7 (C-11) ppm. ESI-MS m/z 435 [M + H]^+^.

##### Lycorine 2-yl (4-methoxylbenzyl)carbamate (**11**)

Obtained from the reaction between lycorine (50 mg, 0.17 mmol, 1 equiv.), CDI (84 mg, 0.52 mmol, 3 equiv.), and 4-methoxybenzylamine (75 µL,0.52 mmol, 3 equiv.). The residue was purified by column chromatography (silica gel CH_2_Cl_2_/MeOH, 100 to 97:3) to afford 40 mg of compound **11** (0.088 mmol, yield 51%) as an amorphous powder. ^1^H NMR (300 MHz, CDCl_3_) δ 7.20 (1H, *bd*, *J* = 8.5 Hz, H-5′, H-9′), 6.85 (1H, *bd*, *J* = 8.2 Hz, H-6′, H-8′), 6.79 (1H, *s*, H-10), 6.58 (1H, *s*, H-7), 5.90 (2H, *dd*, *J* = 8.6, 1.4 Hz, -OCH_2_O), 5.49 (1H *bs*, H-3), 5.29 (1H, *t*, *J* = 5.9 Hz, -NH), 5.22 (1H, *bs*, H-2), 4.51 (1H, *bs*, H-1), 4.28 (1H, *bd*, *J* = 5.8 Hz, H-3′), 4.09 (1H, *d*, *J* = 14.1 Hz, H-6β), 3.78 (3H, *s*, -OCH_3_), 3.61 (1H, *d*, *J* = 14.0 Hz, H-6α), 3.36–3.28 (1H, *m*, H-12β), 2.95 (1H, *d*, *J* = 9.8 Hz, H-4a), 2.67 * (3H, *bd*, *J* = 10.9 Hz, H-10b, H-11), 2.54 * (1H, *m*, H-12α) ppm, * overlapped signals. ^13^C NMR (75 MHz, CDCl_3_) δ 159.0 (-OCH3), 156.0 (C-1′), 146.8 (C-8), 146.3 (C-9), 144.5 (C-4), 130.4 (C-10a, C-4′), 128.9 (C-5′, C-9′), 127.6 (C-6a), 114.8 (C-3), 114.0 (C-6′, C-8′), 107.6 (C-7), 104.9 (C-10), 101.0 (-OCH_2_O), 74.1 (C-2), 68.8 (C-1), 60.6 (C-4a), 56.4 (C-6), 55.2 (-OCH_3_), 53.7 (C-12), 44.6 (C-3′), 41.0 (C-10b) ppm. ESI-MS m/z 451 [M + H]^+^.

##### Lycorine 2-yl benzyl(methyl)carbamate (**12**)

Obtained from the reaction between lycorine (50 mg, 0.17 mmol, 1 equiv.), CDI (84 mg, 0.52 mmol, 3 equiv.), and *N*-benzylmethylamine (68.7 µL, 0.52 mmol, 3 equiv.). The residue was purified by column chromatography (silica gel CH_2_Cl_2_/MeOH, 100 to 97:3) to afford 42 mg of compound **12** (0.096 mmol, yield 54.4%) as an amorphous powder. ^1^H NMR (300 MHz, CDCl_3_) δ 7.31 (4H, *m*, H-Ar), 7.15 (1H, *d*, *J* = 7.2 Hz, H-Ar), 6.81 (1H, *s*, H-10), 6.57 (1H, *s*, H-7), 5.91 (2H, *dd*, *J* = 11.3, 1.5 Hz, -OCH_2_O), 5.48 (1H, *bs*, H-2), 5.30 (1H, *bs*, H-3), 4.56 * (1H, *bs*, H-1), 4.49 * (1H, *d*, *J* = 12.8 Hz, H-3′), 4.40 * (1H, *d*, *J* = 11.9 Hz, H-3′), 4.12 (1H, *dd*, *J* = 13.5, 0.3 Hz, H-6β), 3.55 (1H, *d*, *J* = 13.1 Hz, H-6α), 3.35 (1H, *m*, H-12β), 2.94 (3H, *s*, -NCH_3_), 2.84 (1H, *d*, *J* = 10.6 Hz, H-4a), 2.79 (1H, *s*, H-10a), 2.63 (2H, *d*, *J* = 8.2 Hz, H-11), 2.39 (1H, *q*, *J* = 8.8 Hz, H-12α) ppm. ^13^C NMR (75 MHz, CDCl_3_) δ 146.6 (C-8), 146.3 (C-9), 145.7 (C-4), 137.4 (C-3′), 129.9 (C-10a), 128.7 (C-5′, C-7′), 127.9 (C-8′), 127.6 (C-6a), 127.5 (C-4′), 127.4 (C-6′), 114.4 (C-3), 107.6 (C-7), 104.9 (C-10), 101.0 (-OCH_2_O), 75.0 (C-2), 69.4 (C-1), 60.8 (C-4a), 57.0 (C-6), 53.8 (C-12), 41.7 (C-10b), 34.7 (-NCH_3_), 28.8 (C-11) ppm. ESI-MS m/z 435 [M + H]^+^.

##### Lycorine 2-yl 3,4-dihydroisoquinoline-2(1H)-carboxylate (**13**)

Obtained from the reaction between lycorine (56 mg, 0.19 mmol, 1 equiv.), CDI (94.8 mg, 0.58 mmol, 3 equiv.), and 1,2,3,4-tetrahydroisoquinoline (74.2 µL, 0.58 mmol, 3 equiv.). The residue was purified by column chromatography (silica gel CH_2_Cl_2_/MeOH, 100:0 to 97:3) to afford 44 mg of compound **13** (0.098 mmol, yield 50.5%) as an amorphous powder. ^1^H NMR (300 MHz, CDCl_3_) δ 7.120–7.10 * (4H, *m*, H-Ar), 6.85 (1H, *s*, H-10), 6.59 (1H, *s*, H-7), 5.89 (2H, *dd*, *J* = 6.5, 1.5 Hz, -OCH_2_O), 5.50 (1H, *bs*, H-2), 5.32 (1H, *s*, H-3), 4.64 (1H, *bs*, H-1), 4.59 (1H, *bs*, H-3′), 4.55 (1H, *bs*, H-3′), 4.16 (1H, *d*, *J* = 14.1 Hz, H-6β), 3.75 -3.69 * (2H, *m*, H-7′), 3.59 * (1H, *d*, *J* = 13.5 Hz, H-6α), 3.41 -3.33(1H, *m*, H-12β), 2.91 * (1H, *d*, *J* = 10.5 Hz, H-4a), 2.87 -2.78 * (2H, *m*, H-6′), 2.74 * (1H, *bd*, *J* = 10.7 Hz, H-10b), 2.71–2.63 * (2H, *m*, H-11), 2.47 (1H, *q*, *J* = 8.8 Hz, H-12α) ppm, * overlapped signals. ^13^C NMR (75 MHz, CDCl_3_) δ 155.1 (C-1′), 146.6(C-8), 146.2 (C-9), 145.3 (C-4), 134.4 (4′, C-5′), 129.5 (C-10a), 128.7 (C-11′), 127.6 (C-6a), 126.5 (C-8′), 126.3 (C-9′, C-10′), 114.4 (C-3), 107.5 (C-7), 104.9 (C-10), 100.9 (-OCH_2_O), 74.6 (C-2), 69.0 (C-1), 60.7 (C-4a), 56.8 (C-6), 53.7 (C-12), 45.3 (C-3′, C-7′), 41.5 (C-10b), 28.7 (C-11) ppm. ESI-MS m/z 447 [M + H]^+^.

##### Lycorine 1,2-diyl bis(pyrrolidine-1-carboxylate) (**14**) and lycorine 2-yl pyrrolidine-1-carboxylate (**15**)

Obtained from the reaction between lycorine (59 mg, 0.20 mmol, 1 equiv.), CDI (99.8 mg, 0.61 mmol, 3 equiv.), and pyrrolidine (51 µL, 0.61 mmol, 3 equiv.). The residue was purified by column chromatography (silica gel CH_2_Cl_2_/MeOH, 100 to 97:3) to afford 11 mg of compound **14** (0.022 mmol, yield 11%) and 64 mg of compound **15** (0.16 mmol, yield 81%) as amorphous powders. 

Lycorine 1,2-diyl bis(pyrrolidine-1-carboxylate) (**14**): ^1^H NMR (300 MHz, CDCl_3_) δ 6.90 (1H, *d*, *J* = 0.9 Hz, H-10), 6.55 (1H, *s*, H-7), 5.90 (2H, *dd*, *J* = 6.6, 1.4 Hz, -OCH_2_O), 5.68 (1H, *bs*, H-1), 5.61 (1H, *bs*, H-3), 5.25–5.20 (1H, *m*, H-2), 4.14 (1H, *d*, *J* = 14.0 Hz, H-6β), 3.50 (1H, *dd*, *J* = 14.1, 2.0 Hz, Hz, H-6α), 3.44–3.22 * (7H, *m*, H-3′, H-6′, H-3′′, H-6′′), 3.20 -3.10 (1H, *m*, H-12β), 3.04–2.96 (1H, *m*, H-3′), 2.85 (1H, *d*, *J* = 10.6 Hz, H-4a), 2.75 (1H, *bd*, *J* = 10.5 Hz, H-10b), 2.63 (2H, *m*, H-11), 2.38 (1H, *q*, *J* = 8.8 Hz, H-12α), 1.86 -1.80 * (4H, *m*, 4′, H-5′), 1.78–1.70 * (4H, *dq*, *J* = 8.2, 5.2 Hz, H-4′′, H-5′′) ppm, * overlapped signals. ^13^C NMR (75 MHz, CDCl_3_) δ 153.8 (C-1′′), 153.8 (C-1′), 146.3 (C-8), 146.1 (C-9), 144.8 (C-4), 129.2 (C-10a), 127.3 (C-4a), 115.2 (C-3), 107.1 (C-7), 105.5 (C-10, 100.8 (-OCH_2_O), 71.5 (C-2), 70.1 (C-1), 61.5 (C-4a), 56.9 (C-6), 53.7 (C-12), 46.1 (C-3′, C-3′′), 45.7 (C-6′), 45.5 (C-6′′), 40.8 (C-10b), 28.6 (C-11), 25.6 (C-4′), 25.4 (C-4′′), 24.9 (C-5′), 24.8 (C-5′′) ppm. ESI-MS m/z 482 [M + H]^+^.

Lycorine 2-yl pyrrolidine-1-carboxylate (**15**): ^1^H NMR (300 MHz, CDCl_3_) δ 6.85 (1H, *s*, H-10), 6.57 (1H, *s*, H-7), 5.90 (2H, *dd*, *J* = 7.1, 1.2 Hz, -OCH_2_O), 5.48 (1H, *bs*, H-3), 5.26 (1H, *bs*, H-2), 4.57 (1H, *bs*, H-1), 4.14 (1H, *d*, *J* = 14.1 Hz, H-6β), 3.53 (1H, *dd*, *J* = 14.2, 2.1 Hz, H-6α), 3.44–3.38 * (2H, *m*, H-6′), 3.35 * (1H, *dd*, *J* = 9.0, 4.5 Hz, H-12β), 3.30 -3.27 (2H, *m*, H-3′), 2.82 (1H, *d*, *J* = 10.4 Hz, H-4a), 2.68 * (1H, d, *J* = 10.0 Hz, H-10a), 2.63 * (2H, *m*, H-11), 2.38 (1H, *dd*, *J* = 8.6, 5.6 Hz, H-12α), 1.90–1.80 (4H, *m*, H-4′, H-5′). ^13^C NMR (75 MHz, CDCl_3_) δ 154.8 (C-1′), 146.6 (C-8), 146.3 (C-9), 145.4 (C-4), 130.0 (C-10a), 127.9 (C-6a), 114.6 (C-3), 100.0 (-OCH_2_O), 74.4 (C-2), 69.5 (C-1), 60.8 (C-4a), 57. 1 (C-6), 53.8 (C-12), 46.3 (C-3′), 46.0 (C-6′), 41.7 (C-10b), 28.8 (C-11), 25.7 (C-4′), 25.0 (5′) ppm. ESI-MS m/z 385 [M + H]^+^.

##### Lycorine 1,2-diyl bis(piperidine-1-carboxylate) (**16**) and lycorine 2-yl piperidine-1-carboxylate (**17**)

Obtained from the reaction between lycorine (63 mg, 0.21 mmol, 1 equiv.), CDI (106 mg, 0.66 mmol, 3 equiv.), and piperidine (65 µL, 0.66 mmol, 3 equiv.). The residue was purified by column chromatography (silica gel CH_2_Cl_2_/MeOH, 100 to 97:3) to afford 22 mg of compound **16** (0.043 mmol, yield 19.6%) and 42 mg of compound **17** (0.10 mmol, yield 48%) as amorphous powders. 

Lycorine 1,2-diyl bis(piperidine-1-carboxylate) (**16**): ^1^H NMR (300 MHz, CDCl_3_) δ 6.89 (1H, *s*, H-10), 6.56 (1H, *s*, H-7), 5.89 (2H, *s*, -OCH_2_O), 5.70 (1H, *bs*, H-1), 5.61 (1H, *bs*, H-2), 5.22 (1H, *bs*, H-3), 4.15 (1H, *d*, *J* = 14.1 Hz, H-6β), 3.51 (1H, *dd*, *J* = 14.1, 2.0 Hz, H-6α), 3.44–3.02 * (9H, *m*, H-12β, H-3′, H-3′′, H-7′, H-7′′), 2.85 (1H, *d*, *J* = 10.3 Hz, H-4a), 2.72 * (1H, *d*, *J* = 10.4 Hz, H-10b), 2.64 * (2H, *m*, H-11), 2.40 (1H, *q*, *J* = 8.7 Hz, H-12α), 1.60–1.30 * (12H, *m*, H-4′, H-4′′, H-5′, H5′′, H-6′, H-6′′) ppm, * overlapped signals. ^13^C NMR (75 MHz, CDCl_3_) δ 154.4 (C-1′′), 154.2 (C-1′), 146.3 (C-8), 146.1 (C-9), 144.7 (C-4), 129.3 (C-10a), 127.2 (C-6a), 115.2 (C-3), 107.1 (C-7), 105.6 (C-10), 100.8 (-OCH_2_O), 71.7 (C-2), 70.3 (C-1), 61.6 (C-4a), 56.9 (C-6), 53.7 (C-12), 45.0 (C-3′, C-3′′), 44.8 (C-7′, C-7′′), 40.9 (C-10b), 28.6 (C-11), 25.4 (C-4′, C-4′′, C-6′,C-6′′), 24.3 (C-5′, C-5′′) ppm. ESI-MS m/z 510 [M + H]^+^. 

Lycorine 2-yl piperidine-1-carboxylate (**17**): ^1^H NMR (300 MHz, CDCl_3_) δ 6.84 (1H, *d*, *J* = 1.0 Hz, H-10), 6.58 (1H, *s*, H-7), 5.91 (2H, *dd*, *J* = 7.0, 4.6 Hz, -OCH_2_O), 5.47 (1H, *bs*, H-3), 5.23 (1H, *bs*, H-2), 4.55 (1H, H-1), 4.15 (1H, *d*, *J* = 14.1 Hz, H-6β), 3.54 (1H, *d*, *J* = 13.6 Hz, H-6α), 3.46–3.30 * (5H, *m*, H-12β, H-3′, H-7′), 2.82 (1H, *d*, *J* = 10.7 Hz, H-4a), 2.69 * (1H, *bd*, *J* = 10.4 Hz, H-10b), 2.66–2.60 * (2H, *m*, H-11), 2.39 (1H, *q*, *J* = 8.8 Hz, H-12α), 1.61–1.41 * (6H, *m*, H-4′, H-5′, H-6′) ppm, * overlapped signals. ^13^C NMR (75 MHz, CDCl_3_) δ 155.0 (C-1′), 146.5 (C-8), 146.2 (C-9), 145.3 (C-4), 129.8 (10a), 127.6 (C-6a), 114.4 (C-3), 107.5 (C-7), 104.2 (C-10), 100.9 (-OCH_2_O), 74.5 (C-2), 69.3 (C-1), 60.7 (C-4a), 56.9 (C-6), 53.7 (C-12), 44.8 (C-3′, C-7′), 41.6 (C-10b), 28.6 (C-11), 25.6 (C-4′, C-6′), 24.3 (C-5′) ppm. ESI-MS m/z 399 [M + H]^+^

##### Lycorine 1,2-diyl bis [4-(tert-butyl)benzyl)carbamate] (**18**) and lycorine 2-yl [4-(tert-butyl)benzyl]carbamate (**19**)

Obtained from the reaction between lycorine (50 mg, 0.17 mmol, 1 equiv.), CDI (85 mg, 0.52 mmol, 3 equiv.), and *tert*-butylbenzylamine (92 µL, 0.52 mmol, 3 equiv.). The residue was purified by column chromatography (silica gel CH_2_Cl_2_/MeOH, 100 to 97:3) to afford 33 mg of compound **18** (0.049 mmol, yield 28.4%) and 47 mg of compound **19** (0.098 mmol, yield 56.6%) as amorphous powders. 

Lycorine 1,2-diyl bis[(4-(*tert*-butyl)benzyl)carbamate] (**18**): ^1^H NMR (300 MHz, CDCl_3_) δ 7.36ª (2H, *d*, *J* = 8.3 Hz, H-6′, H-8′), 7.31ª (2H, *d*, *J* = 8.1 Hz, H-5′′, H-9′′), 7.24–7.16 *ª (2H, *m*, H-5′, H-9′), 7.11ª (2H, *d*, *J* = 8.1 Hz, H-6′′,H-8′′), 6.87 (1H, *s*, H-10), 6.56 (1H, *s*, H-7), 5.92 (2H, *s*, -OCH_2_O), 5.72 (1H, *bs*, H-1), 5.62 (1H, *bs*, H-2), 5.31 (1H, *bs*, H-3), 5.12 (1H, *bs*, -NH), 4.88 (1H, *bs*, -NH), 4.37 (1H, *bt*, *J* = 5.2 Hz, H-3′), 4.28 (2H, bt, *J* = 6.1 Hz, H-3′′), 4.08 (1H, *d*, *J* = 13.7 Hz, H-6β), 3.53 (1H, *d*, *J* = 13.8 Hz, H-6α), 3.36–3.26 (1H, *m*, H-12β), 2.80 (2H, *bs*, H-4a, H-10b), 2.62 (2H, *bs*, H-11), 2.40 (1H, *bs*, H-12α), 1.31ª (9H, *s*, C(CH_3_)_3_), 1.29ª (9H, *s*, C(CH_3_)_3_), ppm, ª signals could be changed, * overlapped signals. ^13^C NMR (75 MHz, CDCl_3_) δ 155.5 (C-1′′), 155.4 (C-1′), 150.6 (C-7′, C-7′′), 146.7 (C-8), 146.4 (C-9), 144.9 (C-4), 135.4 (C-4′), 135.3 (C-4′′), 127.5 (C-9′, 9′′), 127.3 (C-5′, C-5′′), 127.1 (C-6a), 125.7 (C-8′, C-8′′), 125.6 (C-6′, C-6′′), 115.1 (C-3), 107.4 (C-7), 105.5 (C-10), 101.0 (-OCH_2_O), 71.5 (C-2), 70.2 (C-1), 56.8 (C-6), 53.8 (C-12), 45.0 (C-3′′), 44.9 (C-3′), 34.6 (C(CH_3_)_3_), 34.5 (C(CH_3_)_3_), 31.4 (C(CH_3_)_3_), 28.8 (C-11) ppm. ESI-MS m/z 666 [M + H]^+^.

Lycorine 2-yl [4-(*tert*-butyl)benzyl]carbamate (**19**): ^1^H NMR (300 MHz, CDCl_3_) δ 7.36 (2H, *bd*, *J* = 8.0 Hz, H-6′, H-8′), 7.22 (2H, *bd*, *J* = 8.0 Hz, H-5′, H-9′), 6.78 (1H, *s*, H-10), 6.57 (1H, *s*, H-7), 5.89 (2H, *dd*, *J* = 7.2, 1.5 Hz, -OCH_2_O), 5.48 (1H, *bs*, H-3), 5.34–5.28 (1H, *m*, -NH), 5.24 (1H, *bs*, H-2), 4.51 (1H, *bs*, H-1), 4.33 (2H, *bd*, *J* = 5.9 Hz, H-3′), 4.11 (1H, *d*, *J* = 14.0 Hz, H-6β), 3.51 (1H, *d*, *J* = 14.1 Hz, H-6α), 3.32 (1H, *dt*, *J* = 9.0, 4.6 Hz, H-12β), 2.82 (1H, *d*, *J* = 10.6 Hz, H-4a), 2.65 * (1H, *bs*, H-10b), 2.61 * (1H, *bs*, H-11), 2.37 (1H, *q*, *J* = 8.7 Hz, H-12α), 1.31 (9H, *s*, C(CH_3_)_3_) ppm, * signals overlapped. ^13^C NMR (75 MHz, CDCl_3_) δ 156.2 (C-1′), 150.6 (C-7′), 146.6 (C-8), 146.3 (C-9), 145.7 (C-4), 135.4 (C-4′), 129.9 (C-10a), 127.7 (C-6a), 127.4 (C-5′, C-9′), 125.7 (C-6′, C-8′), 114.3 (C-3), 107.7 (C-7), 104.9 (C-10), 101.0 (-OCH_2_O), 74.5 (C-2), 69.3 (C-1), 60.8 (C-4a), 57.0 (C-6), 53.8 (C-12), 44.9 (C-3′), 41.6 (C-10b), 34.6 (C(CH3)3), 31.3 (C(CH_3_)_3_), 28.8 (C-11) ppm. ESI-MS m/z 477 [M + H]^+^.

##### Lycorine 1,2-diyl bis[(2-fluorobenzyl)carbamate] (**20**) and lycorine 2-yl (2-fluorobenzyl)carbamate (**21**)

Obtained from the reaction between lycorine (60 mg, 0.20 mmol, 1 equiv.), CDI (101.5 mg, 0.62 mmol, 3 Equiv.), and 2-fluorobenzylamine (71.6 µL, 0.62 mmol, 3 equiv.). The residue was purified by column chromatography (silica gel CH_2_Cl_2_/MeOH, 100 to 98:2) to afford 56 mg of compound **20** (0.094 mmol, yield 45.4%) and 58 mg of compound **21** (0.13 mmol, yield 63.3%) as amorphous powders. 

Lycorine 1,2-diyl bis[(2-fluorobenzyl)carbamate] (**20**): ^1^H NMR (300 MHz, CDCl_3_) δ 7.36 (1H, *t*, *J* = 7.5 Hz, H-Ar), 7.25–7.17 (3H, *m*, H-Ar), 7.13 (1H, *bd*, *J* = 7.4 Hz, H-Ar), 7.09–6.93 (3H, *m*, H-Ar), 6.83 (1H, *s*, H-10), 6.54 (1H, *s*, H-7), 5.90 (2H, *s*, -OCH_2_O), 5.68 (1H, *bs*, H-1), 5.58 (1H, *bs*, H-2), 5.26 (1H, *bs*, H-3), 5.13 (1H, *bs*, -NH), 4.93 (1H, *t*, *J* = 6.3 Hz, -NH), 4.43 (2H, *t*, *J* = 5.8 Hz, H-3′), 4.36 (2H, *bd*, *J* = 6.2 Hz, H-3′′), 4.13 (1H, *d*, *J* = 14.0 Hz, H-6β), 3.49 (1H, *d*, *J* = 14.1 Hz, H-6α), 3.38 -3.29 (1H, *m*, H-12β), 2.80 (1H, *d*, *J* = 10.5 Hz, H-4a), 2.72 (1H, *d*, *J* = 10.5 Hz, H-10b), 2.61 (2H, *bs*, H-11), 2.41–2.29 (1H, *m*, H-12α) ppm. ^13^C NMR (75 MHz, CDCl_3_) δ 162.1 (C-5′, C-5′′), 155.4 (C-1′′), 155.2 (C-1′), 146.4 (C-8), 146.2 (C-9), 145.4 (C-4), 130.0 (C-7′′, C-9′′), 129.3(C-10a), 129.2 (C-7′, C-9′), 126.8 (C-6a), 125.6 (C-4′, C-4′′), 124.3 (C-8′, C-8′′), 115.3 (C-6′), 115.2 (C-6′′), 114.5 (C-3), 107.2 (C-7), 105.3 (C-10), 100.9 (-OCH_2_O), 71.6 (C-2), 70.3 (C-1), 61.3 (C-4a), 56.9 (C-6), 53.6 (C-12), 40.5 (C-10b), 39.7 (C-3′, C-3′′), 28.8 (C-11) ppm. ESI-MS m/z 590 [M + H]^+^. 

Lycorine 2-yl (2-fluorobenzyl)carbamate (**21**): ^1^H NMR (300 MHz, CDCl_3_) δ 7.32 (1H, *bt*, *J* = 7.9 Hz, H-7′), 7.23 (1H, *bd*, *J* = 6.2 Hz, H-9′), 7.11 (1H, *bd*, *J* = 7.4 Hz, H-8′), 7.04 (1H, *t*, *J* = 8.6 Hz, H-6′), 6.74 (1H, *s*, H-10), 6.56 (1H, *s*, H-7), 5.89 (2H, *dd*, *J* = 8.7, 1.5 Hz, -OCH_2_O), 5.51 (1H, *bs*, NH), 5.46 (1H, *bs*, H-2), 5.20 (1H, *bs*, H-3), 4.44 (1H, *bs*, H-1), 4.39 (2H, *bd*, *J* = 6.1 Hz, H-3′), 4.10 (1H, *dd*, *J* = 13.9, 2.7 Hz, H-6β), 3.49 (1H, *d*, *J* = 14.1 Hz, H-6α), 3.30 (1H, *dt*, *J* = 9.3, 4.7 Hz, H-12β), 2.80 (1H, *d*, *J* = 10.6 Hz, H-4a), 2.63 * (1H, *bs*, H-10b), 2.60 * (2H, *s*, H-11), 2.36 (1H, *m*, H-12α) ppm, * overlapped signals. ^13^C NMR (75 MHz, CDCl_3_) δ 162.5 (C-5′), 159.3 (C-5′), 156.1 (C-1′), 146.5 (C-8), 146.2 (C-9), 145.6 (C-4), 129.9 (C-7′), 129.7 (C-10a), 129.4 (9′), 129.2 (C-9′), 127.6 (C-6a), 125.5 (C-4′), 125.3 (C-4′), 124.3 (C-8′), 124.2 (C-8′), 115.5 (C-6′), 115.2 (C-6′), 114.1 (C-3), 107.5 (C-7), 104.8 (C-10), 100.9 (-OCH_2_O), 74.5 (C-2), 69.0 (C-1), 60.7 (C-4a), 56.9 (C-6), 53.6 (C-12), 41.5 (C-10b), 39.1 (C-3′), 28.6 (C-11). ESI-MS m/z 439 [M + H]^+^.).

##### Lycorine 1,2-diyl bis[(3-(trifluoromethyl)benzyl)carbamate] (**22**)

Obtained from the reaction between lycorine (60 mg, 0.20 mmol, 1 equiv.), CDI (101.5 mg, 062 mmol, 3 equiv.), and 3-(trifluoromethyl) benzylamine (152.6 µL, 0.62 mmol, 3 equiv.). The residue was purified by column chromatography (silica gel CH_2_Cl_2_/MeOH, 100 to 98:2) to afford 52 mg of compound **22** (0.075 mmol, yield 36.1%) as an amorphous powder. ^1^H NMR (300 MHz, CDCl_3_) δ 7.56–7.47 * (4H, *m*, H-Ar), 7.41 * (4H, *bd*, *J* = 8.3 Hz, H-Ar), 6.82 (1H, *s*, H-10), 6.54 (1H, *s*, H-7), 5.90 (2H, *s*, OCH_2_O), 5.71 (1H, *s*, -NH), 5.59 (1H, *s*, H-3), 5.32 (1H, *s*, H-2), 5.03 (1H, *s*, H-1), 4.39 (4H, *m*, H-3′, H-3′′), 4.11 (1H, *d*, *J* = 14.2 Hz, H-6β), 3.48 (1H, *d*, *J* = 14.1 Hz, H-6α), 3.33 (1H, *m*, H-12β), 2.82 (1H, *d*, *J* = 10.6 Hz, H-4a), 2.72 (1H, *d*, *J* = 10.3 Hz, H-10b), 2.61 (2H, *s*, H-11), 2.35 (1H, *q*, *J* = 8.7 Hz, H-12α) ppm, ** overlapped signals. ^13^C NMR (75 MHz, CDCl_3_) δ 155.6 (C-1′), 155.5 (C-1′′), 146.6 (C-8), 146.4 (C-9), 145.8 (C-4), 139.6 (C-4′), 139.3 (C-4′′), 131.0 (C-5′/9′), 130.9 (C-5′′, C-9′′), 129.4 (C-10a), 129.3 (C-5′, C-5′′, C-9′, C-9′′), 126.8 (C-6′, C-7′, C-7′′), 124.4 (C-6′, C-6′′,C-8′, C-8′′), 114.5 (C-3), 107.3 (C-7), 105.3 (C-10), 101.1 (-OCH2O), 71.9 (C-2), 70.7 (C-1), 61.3 (C-4a), 57.1 (C-6), 53.7 (C-12), 44.8 (C-3′, C.3′′), 40.6 (C-10b), 28.8 (C-11) ppm. ESI-MS m/z 690 [M + H]^+^.

##### Lycorine 1,2-diyl bis[(4-fluorophenethyl) carbamate] (**23**) and lycorine 2-yl (4-fluorophenethyl)carbamate (**24**)

Obtained from the reaction between lycorine (63 mg, 0.21 mmol, 1 equiv.), CDI (106.6 mg, 0.65 mmol, 3 equiv.), and 4-fluoro phenethylamine (75.4 µL, 0.65 mmol, 3 equiv.). The residue was purified by column chromatography (silica gel CH_2_Cl_2_/MeOH, 100 to 98:2) to afford 22 mg of compound **23** (0.035 mmol, yield 16.2%) and 24 mg of compound **24** (0.053 mmol, yield 24.1%) as amorphous powders. 

Lycorine 1,2-diyl bis[(4-fluorophenethyl) carbamate] (**23**): ^1^H NMR (300 MHz, CDCl_3_) δ 7.14 (2H, *bt*, *J* = 7.3 Hz, H-6′, H-6′′), 6.99 * (4H, *bt*, *J* = 7.7 Hz, H-7′, H-7′′, H-9′, H-9′′), 6.90 * (2H, *t*, *J* = 8.7 Hz, H-10′, H-10′′), 6.85 * (1H, *s*, H-10), 6.56 (1H, *s*, H-7), 5.89 (2H, *bs*, -OCH_2_O), 5.64 (1H, *bs*, H-1), 5.54 (1H, *bs*, H-2), 5.21 (1H, *bs*, H-3), 4.82 (1H, *t*, *J* = 6.3 Hz, -NH), 4.59 (1H, *t*, *J* = 6.1 Hz, -NH), 4.13 (1H, *d*, *J* = 14.1 Hz, H-6β), 3.48 * (1H, *d*, *J* = 14.7 Hz, H-6α), 3.44–3.23 * (4H, *m*, H-3′, H-3′′), 2.85–2.67 * (6H, *m*, H-4a, H-10a, H-4′, H-4′′), 2.61 * (2H, *bs*, H-11), 2.35 (1H, *q*, *J* = 8.7 Hz, H-12α) ppm, * overlapped signals. ^13^C NMR (75 MHz, CDCl_3_) δ 163.4 (C-8′), 163.3 (C-8′′), 160.1 (C-8′), 160.0 (C-8′′), 155.4 (C-1′), 155.3 (C-1′′), 146.5 (C-8), 146.4 (C-9), 145.5 (C-4), 134.5 (C-5′, C-5′′), 130.4 (C-6′), 130.3 (C-6′′), 130.2 (C-10′, C-10′′), 129.5 (C-10a), 127.0 (C-6a), 115.6 (C-7′, C-7′′), 115.5 (C-7′, C-7′′), 115.4 (C-9′,C-9′′), 115.2 (C-9′, C-9′′), 114.7 (C-4), 107.3 (C-7), 105.5 (C-10), 101.0 (-OCH_2_O), 71.6 (C-2), 70.2 (C-1), 61.3 (C-4a), 57.0 (C-6), 53.8 (C-12), 42.4 (C-3′, C-3′′), 40.6 (C-10b), 35.4 (C-4′), 35.0 (C-4′′), 27.9 (C-11) ppm. ESI-MS m/z 618 [M + H]^+^.

Lycorine 2-yl (4-fluorophenethyl)carbamate (**24**): ^1^H NMR (300 MHz, CDCl_3_) δ 7.13 (2H, *bdd*, *J* = 8.3, 5.4 Hz, H-6′, H-10′), 6.97 (2H, *bt*, *J* = 8.5 Hz, H-7′, H-9′), 6.78 (1H, *s*, H-10), 6.59 (1H, *s*, H-7), 5.89 (2H, *d*, *J* = 7.3 Hz, -OCH2O), 5.45 (1H, *bs*, H-2), 5.19 (1H, *m*, H-3), 4.99 (1H, *t*, *J* = 6.3, Hz, -NH), 4.47 (1H, *bs*, H-1), 4.12 (1H, *d*, *J* = 14.0 Hz, H-6β), 3.51 (1H, *d*, *J* = 14.1 Hz, H-6α), 3.40 * (2H, *bd*, *J* = 6.8 Hz, H-3′), 3.36–3.27 * (1H, *m*, H-12β), 2.84–2.74 (3H, *m*, H-4′, H-4a), 2.62 (3H, *bd*, *J* = 9.1 Hz, H-11, H-10a), 2.37 (1H, *t*, *J* = 8.7 Hz, H-12α) ppm, * overlapped signals.^13^C NMR (75 MHz, CDCl_3_) δ 163.4 (C-8′), 160.1 (C-8′), 156.1 (C-1′), 146.6 (C-8), 146.3 (C-9), 145.8 (C-4), 134.4 (C5′), 130.3 (C-6′), 130.2 (C-10′), 129.9 (C-10a), 127.7 (C-6a), 115.6 (C-7′), 115.4 (C-9′), 114.2 (C-3), 107.7 (C-7), 104.9 (C-10), 101.0 (-OCH_2_O), 74.4 (C-2), 69.2 (C-1), 60.8 (C-4a), 57.1 (C-6), 53.8 (C-12), 42.4 (C-3′), 41.6 (C-10b), 35.4 (C-4′), 28.8 (C-11) ppm. ESI-MS m/z 453[M + H]^+^.

##### Lycorine 1,2-diyl bis[4-(chlorophenethyl)carbamate] (**25**) and lycorine 2-yl (4-(chlorophenethyl) carbamate (**26**)

Obtained from the reaction between lycorine (64 mg, 0.22 mmol, 1 equiv.), CDI (108.3 mg, 0.66 mmol, 3 equiv.), and 2,4-(chlorophenyl) ethylamine (93.6 µL, 0.66 mmol, 3 Equiv.). The residue was purified by column chromatography (silica gel CH2Cl2/MeOH, 100 to 98:2) to afford 48 mg of compound **25** (0.048 mmol, yield 33.1%) and 65 mg of compound **26** (0.13 mmol, yield 62.3%) as amorphous powders. 

Lycorine 1,2-diyl bis[(4-chlorophenethyl)carbamate] (**25**): ^1^H NMR (300 MHz, CDCl_3_) δ 7.31–7.23 (4H, *m*, H-Ar), 7.12 (3H, *dd*, *J* = 8.5, 2.4 Hz, H-Ar), 6.96 (1H, *bd*, *J* = 8.0 Hz, H-Ar), 6.84 (1H, *s*, H-10), 6.57 (1H, *s*, H-7), 5.90 (1H, *s*, -OCH_2_O), 5.63 (1H, *bs*, H-1), 5.53 (1H, *bs*, H-2), 5.21 (1H, *s*, H-3), 4.81 (1H, *bs*, -NH), 4.58 (1H, *t*, *J* = 6.0 Hz, -NH), 4.13 (1H, *d*, *J* = 14.1 Hz, H-6β), 3.56–3.30 * (6H, *m*, H-3′, H-3′′, H-6α, H-12β), 2.85–2.67 * (6H, *m*, H-4′, H-4′′, H-4a, H-10a), 2.61 * (2H, *m*, H-11), 2.35 (1H, *q*, *J* = 8.8 Hz, H-12α) ppm, * overlapped signals. ^13^C NMR (75 MHz, CDCl_3_) δ 155.4 (C-1′′), 155.3 (C-1′), 146.5 (C-8), 146.4 (C-9), 145.5 (C-4), 137.3 (C-5′, C-5′′), 132.4 (C-8′), 132.3 (C-8′′), 130.2 (C-6′, C-6′′, C-10′, C-10′′), 129.5 (C-10a), 128.9 (C-9′), 128.8 (C9′′), 128.7 (C-7′, C-7′′), 127.0 (C-6a), 114.6 (C-4), 107.3 (C-7), 105.5 (C-10), 101.1 (-OCH_2_O), 71.6 (C-2), 70.3 (C-1), 57.0 (C-6), 53.8 (C-12), 42.2 (C-3′, C-3′′), 39.2 (C-4′, C-4′′), 28.6 (C-11) ppm. ESI-MS m/z 650 [M + H]^+^. 

Lycorine 2-yl (4-chlorophenethyl) carbamate (**26**): ^1^H NMR (300 MHz, CDCl_3_) δ 7.26 (2H, *bd*, *J* = 9.0 Hz, H-6′, H-10′), 7.12 (2H, *bd*, *J* = 8.0 Hz, H-7′, H-9′), 6.79 (1H, *d*, *J* = 1.0 Hz, H-10), 6.59 (1H, *s*, H-7), 5.91 (2H, *dd*, *J* = 7.8, 1.3 Hz,-OCH_2_O), 5.45 (1H, *bs*, H-2), 5.20 (1H, *bs*, H-3), 4.84 (1H, *bs*, -NH), 4.50 (1H, *bs*, H-1), 4.13 (1H, *d*, *J* = 14.1 Hz, H-6β), 3.52 (1H, *d*, *J* = 14.2 Hz, H-6α), 3.43 * (2H, *m*, H-3′), 3.33 * (1H, *m*, H-12β), 2.83–2.76 * (3H, *m*, H-4′, H-4a), 2.63 * (3H, *bd*, *J* = 9.3 Hz, H-11, H-10b), 2.37 (1H, *q*, *J* = 8.8 Hz, H-12α) ppm, * overlapped signals.^13^C NMR (75 MHz, CDCl_3_) δ 156.1 (C-1′), 146.7 (C-8), 146.4 (C-9), 145.9 (C-4), 137.2 (C-5′), 132.5 (C-8′), 130.2 (C-6′, C-10′), 130.20 (C-10a), 128.9 (C-7′, C-9′), 127.6 (C-6a), 114.2 (C-4), 107.7 (C-7), 104.8 (C-10), 101.1 (-OCH_2_O), 74.4 (C-2), 69.3 (C-1), 60.8 (C-4a), 57.1 (C-6), 53.8 (C-12), 42.2 (C-3′), 41.6 (C-10b), 38.9 (C-4′), 28.8 (C-11) ppm. ESI-MS m/z 469 [M + H]^+^.

##### Lycorine 1,2-diyl bis[(4-fluorobenzyl)carbamate] (**27**) and Lycorine 2-yl (4-fluorobenzyl)carbamate (**28**)

Obtained from the reaction between lycorine (63 mg, 0.20 mmol, 1 equiv.), CDI (101.5 mg, 0.62 mmol, 3 equiv.), and 4-fluorobenzylamine (70.6 µL, 0.62 mmol, 3 equiv.). The residue was purified by column chromatography (silica gel CH_2_Cl_2_/MeOH, 100 to 98:2) to afford 90 mg of compound **27** (0.15 mmol, yield 73.1%) and 10 mg of compound **28** (0.022 mmol, yield 10.9%) as amorphous powders. 

Lycorine 1,2-diyl bis[(4-fluorobenzyl)carbamate] (**27**): ^1^H NMR (300 MHz, CDCl_3_) δ 7.32–7.21 (2H, *m*, H-Ar), 7.12 (2H, *bt*, *J* = 6.0, 0.2 Hz, H-Ar), 7.03 (2H, *bd*, *J* = 8.8 Hz, H-Ar), 6.96 (2H, *bdd*, *J* = 9.4, 1.6 Hz, H-Ar), 6.82 (1H, *s*, H-10), 6.54 (1H, *s*, H-7), 5.91 (2H, *bs*, -OCH_2_O), 5.69 (1H, *bs*, H-1), 5.58 (1H, *bs*, H-2), 5.29 * (1H, *s*, -NH), 5.24 * (1H, *bs*, H-3), 4.94 (1H, *t*, *J* = 5.8, Hz, -NH), 4.34ª * (2H, *bt*, *J* = 6.0 Hz, H-3′), 4.25ª * (2H, *dd*, *J* = 13.6, 5.8 Hz, H-3′′), 4.11 (1H, *d*, *J* = 13.9 Hz, H-6β), 3.47 (1H, *d*, *J* = 14.1 Hz, H-6α), 3.33 (1H, *dd*, *J* = 9.4, 4.6 Hz, H-12β), 2.79 * (1H, *d*, *J* = 10.5 Hz, H-10b), 2.70 * (1H, *d*, *J* = 10.9 Hz, H-4a), 2.60 (2H, *m*, H-11), 2.34 (1H, *q*, *J* = 8.9 Hz, H-12α) ppm, ª signals could be changed, * overlapped signals.^13^C NMR (75 MHz, CDCl_3_) δ 163.9 (C-7′, C-7′′), 160.7 (C-7′, C-7′′), 155.6 (C-1′), 155.4 (C-1′′), 146.5 (C-8), 146.4 (C-9), 145.6 (C-4), 134.2 (C-4′, C-4′′), 129.5 (C-10a), 129.4 (C-9′), 129.3 (C-5′, C-9′), 129.2 (C-5′′), 126.9 (C-6a), 115.7 (C-8′, C-8′′), 115.6 (C-8′, C-8′′), 115.5 (C-6′, C-6′′), 115.4 (C-6′, C-6′′), 114.6 (C-4), 107.3 (C-7), 105.4 (C-10), 101.1 (-OCH_2_O), 71.8 (C-2), 70.5 (C-1), 61.4 (C-4a), 57.0 (C-6), 53.8 (C-12), 44.6 (C-3′), 44.5 (C-3′′), 40.2 (C-10b), 28.8 (C-11). ESI-MS m/z 590 [M + H]^+^. 

Lycorine 2-yl (4-fluorobenzyl)carbamate (**28**): ^1^H NMR (300 MHz, CDCl_3_) δ 7.25 (2H, *dd*, *J* = 8.3, 5.5 Hz, H-5′, H-9′), 7.01 (2H, *bt*, *J* = 8.5 Hz, H-6′, H-8′), 6.77 (1H, *s*, H-10), 6.57 (1H, *s*, H-7), 5.91 (2H, *dd*, *J* = 8.8, 1.4 Hz, -OCH_2_O), 5.47 (1H, *bs*, H-2), 5.32 (1H, *bs*, -NH), 5.23 (1H, *dt*, *J* = 3.4, 1.7 Hz, H-3), 4.51 (1H, *bs*, H-1), 4.32 (2H, *dd*, *J* = 6.4, 2.5 Hz, H-3′), 4.11 (1H, *d*, *J* = 14.1 Hz, H-6β), 3.51 (1H, *d*, *J* = 14.3 Hz, H-6α), 3.39–3.27 (1H, *m*, H-12β), 2.80 (1H, *bd*, *J* = 10.5 Hz, H-10b), 2.65 * (1H, *bs*, H-4a), 2.65–2.56 * (2H, *m*, H-11), 2.36 (1H, *q*, *J* = 8.8 Hz, H-12α) ppm, * overlapped signals.^13^C NMR (75 MHz, CDCl_3_) δ 163.8 (C-7′),160.5 (C-7′), 155.9 (C-1′), 146.6 (C-8), 146.3 (C-9), 145.8 (C-4), 134.0 (C-4′), 129.8 (C-10a), 129.2 (C-9′), 129.1 (C-5′)), 127.4 (C-6a), 115.6 (C-8′, C-6′), 115.4 (C-6′, C-8′), 114.0 (C-3), 107.6 (C-7), 104.7 (C-10), 100.9 (-OCH_2_O), 74.4 (C-2), 69.1 (C-1), 60.7 (C-4a), 56.9 (C-6), 53.6 (C-12), 44.4 (C-3′), 41.5 (C-10b), 28.7 (C-11) ppm. ESI-MS m/z 439 [M + H]^+^.

##### Lycorine 1,2-diyl bis(propylcarbamate) (**29**) and lycorine 2-yl propylcarbamate (**30**)

Obtained from the reaction between lycorine (60 mg, 0.20 mmol, 1 equiv.), CDI (101.5 mg, 0.62 mmol, 3 equiv.), and 1-propylamine (51.5 µL, 0.62 mmol, 3 equiv.). The residue was purified by column chromatography (silica gel CH_2_Cl_2_/MeOH, 100 to 98:2) to afford 20 mg of compound **29** (0.043 mmol, yield 20.9%) and 45 mg of compound **30** (0.12 mmol, yield 57.8%) as amorphous powders. 

Lycorine 1,2-diyl bis(propylcarbamate) (**29**): ^1^H NMR (300 MHz, CDCl_3_) δ 6.84 (1H, *s*, H-10), 6.55 (1H, *s*, H-7), 5.90 (2H, *dd*, *J* = 2.7, 1.5 Hz, -OCH_2_O), 5.63 (1H, *bs*, H-1), 5.59 (1H, *bs*, H-2), 5.21 (1H, *s*, H-3), 4.74 (1H, *t*, *J* = 5.9 Hz, -NH), 4.57 (1H, *t*, *J* = 6.3 Hz, -NH), 4.14 (1H, *d*, *J* = 14.1 Hz, H-6β), 3.50 (1H, *d*, *J* = 14.1 Hz, H-6α), 3.34 (1H, *dt*, *J* = 9.3, 4.8 Hz, H-12β), 3.12 (4H, *ddt*, *J* = 18.9, 9.4, 6.7 Hz, H-3′, H-3′′), 2.79 (1H, *d*, *J* = 10.5 Hz, H-10b), 2.72 (1H, *bd*, *J* = 10.6 Hz, H-4a), 2.62 (2H, *bs*, H-11), 2.37 (1H, *q*, *J* = 8.7 Hz, H-12α), 1.59–1.37 (4H, *m*, H-4′, H-4′′), 0.91 ª (3H, *t*, *J* = 7.4 Hz, H-5′), 0.84ª (3H, *t*, *J* = 7.4 Hz, H-5′′) ppm. ^13^C NMR (75 MHz, CDCl_3_) δ 155.5 (C-1′), 146.6 (C-8), 146.3 (C-9), 145.3 (C-4), 129.4 (C-10a), 127.2 (C-6a), 115.0 (C-3), 107.3 (C-7), 105.5 (C-10), 101.0 (-OCH_2_O), 71.5 (C-2), 70.1 (C-1), 61.4 (C-4a), 57.1 (C-6), 53.8 (C-12), 42.9 (C-3′), 40.6 (C-10b), 28.8 (C-11), 23.3 (C-4′), 23.1 (C-4′′), 11.2 (C-5′), 11.1 (C-5′′) ppm, ª signals could be changed. ESI-MS m/z 458 [M + H]^+^. 

Lycorine 2-yl propylcarbamate (**30**): ^1^H NMR (300 MHz, CDCl_3_) δ 6.78 (1H, *d*, *J* = 0.9 Hz, H-10), 6.58 (1H, *s*, H-7), 5.90 (2H, *dd*, *J* = 7.8, 1.5 Hz, -OCH_2_O), 5.47 (1H, *bs*, H-2), 5.18 (1H, *bs*, H-3), 4.99 (1H, *t*, *J* = 5.9 Hz, -NH), 4.47 (1H, *bs*, H-1), 4.11 (1H, *d*, *J* = 13.9 Hz, H-6β), 3.51 (1H, *d*, *J* = 14.5 Hz, H-6α), 3.31 (1H, *dt*, *J* = 9.0, 3.6 Hz, H-12β), 3.14 (2H, *dq*, *J* = 13.5, 6.7 Hz, H-3′), 2.81 (1H, *d*, *J* = 10.6 Hz, H-4a), 2.62 * (3H, *bd*, *J* = 9.7 Hz, H-4a, H-11), 2.35 (1H, *q*, *J* = 8.8 Hz, H-12α), 1.51 (3H, *p*, *J* = 7.3 Hz, H-4′), 0.92 (3H, *t*, *J* = 7.4 Hz, H-5′) ppm, * overlapped signals. ^13^C NMR (75 MHz, CDCl_3_) δ 156.2 (C-1′), 146.6 (C-8), 146.3 (C-9), 145.6 (C-4), 129.8 (C-10a), 127.8 (C-6a), 114.4 (C-3), 107.6 (C-7), 104.9 (C-10), 101.0 (-OCH_2_O), 74.2 (C-2), 69.1 (C-1), 60.8 (C-4ª), 57.1 (C-6), 53.8 (C-12), 42.9 (C-3′), 41.6 (C-10b), 28.7 (C-11), 23.2 (C-4′), 11.2 (C-5′) ppm. ESI-MS m/z 373 [M + H]^+^.

##### Lycorine 1,2-diyl bis[(4-methoxybenzyl)(methyl)carbamate] (**31**)

Obtained from the reaction between lycorine (64 mg, 0.22 mmol, 1 equiv.), CDI (108.3 mg, 0.66 mmol, 3 equiv.), and 4-methoxy-*N*-methylbenzylamine (100.2 µL, 0.66 mmol, 3 equiv.). The residue was purified by column chromatography (silica gel CH_2_Cl_2_/MeOH, 100 to 98:2) to afford 44 mg of compound **31** (0.094 mmol, yield 42.5%) as an amorphous powder. ^1^H NMR (300 MHz, CDCl_3_) δ 7.20 (1H, *bd*, *J* = 8.2 Hz, H-Ar), 7.08 (1H, *bd*, *J* = 8.0 Hz, H-Ar), 6.89–6.81 (2H, *m*, H-Ar 6.79 (1H, *s*, H-10), 6.57 (1H, *s*, H-7), 5.90 (2H, *dd*, *J* = 10.0, 1.4 Hz, -OCH_2_O), 5.48 (1H, *bs*, H-2), 5.29 (1H, *bs*, H-3), 4.50 (1H, *bs*, H-1), 4.37 (1H, *d*, *J* = 15.0 Hz, H-2′), 4.27 (1H, *d*, *J* = 15.4 Hz, H-3′), 4.13 (1H, *d*, *J* = 14.1 Hz, H-6β), 3.78 (3H, *s*, -OCH_3_), 3.53 (1H, *d*, *J* = 14.1 Hz, H-6α), 3.39–3.31 (1H, *m*, H-12β), 2.91 (3H, *s*, -NCH_3_), 2.82 (1H, *d*, *J* = 10.5 Hz, H-4a), 2.69–2.59 * (3H, *m*, H-11, H-10b), 2.38 (1H, *q*, *J* = 8.8 Hz, H-12α) ppm, overlapped signals. ^13^C NMR (75 MHz, CDCl_3_) δ 159.3 (C-6′), 156.6 (C-1′), 146.6 (C-8), 146.3 (C-9), 145.7 (C-4), 129.9 (C-3′), 129.5 (C-10a), 129.4 (C-8′), 128.8 (C-4′), 127.7 (C-6a), 114.4 (C-3), 107.6 (C-7), 105.0 (C-10), 101.0 (-OCH_2_O), 74.9 (C-2), 69.4 (C-1), 60.8 (C-4a), 57.1 (C-6), 55.3 (-OCH_3_), 53.8 (C-12), 51.9 (C-2′), 41.8 (C-10b), 34.5 (-NCH_3_), 28.8 (C-11) ppm. ESI-MS m/z 465 [M + H]^+^.

##### Lycorine 1,2-diyl bis(benzylcarbamate) (**32**)

Obtained from the reaction between lycorine (70 mg, 024 mmol, 1 equiv.), CDI (118.5 mg, 0.73 mmol, 3 equiv.), and benzylamine (113 µL, 0.73 mmol, 3 equiv.). The residue was purified by column chromatography (silica gel CH_2_Cl_2_/MeOH, 100 to 98:2) to afford 44 mg of compound **32** (0.079 mmol, yield 42.5%) as an amorphous powder. ^1^H NMR (300 MHz, CDCl_3_) δ 7.37–7.21 (8H, *m*, H-Ar), 7.19–7.10 (2H, *m*, H-Ar), 6.87 (1H, *s*, H-10), 6.54 (1H, *s*, H-7), 5.91 (2H. *s*, -OCH_2_O), 5.72 (1H, *bs*, H-1), 5.60 (1H, *bs*, H-2), 5.31 (1H, *bs*, H-3), 5.21 (1H, *t*, *J* = 6.4 Hz, -NH), 4.93 (1H, *t*, *J* = 6.0 Hz, -NH), 4.38 (2H, *dt*, *J* = 11.9, 6.2 Hz, H-3′, H-3′′), 4.30 (2H, *dt*, *J* = 13.6, 6.6 Hz, H-3′, H-3′′), 4.11 (1H, *d*, *J* = 14.1 Hz, H-6β), 3.47 (1H, d, *J* = 14.8 Hz, H-6α), 3.33 (1H, *dt*, *J* = 9.4, 4.9 Hz, H-12β), 2.81 (1H, *d*, *J* = 10.5 Hz, H-4a), 2.71 (1H, *bd*, *J* = 10.7 Hz, H-10b), 2.60 (2H, *m*, H-11), 2.41–2.29 (1H, *m*, H-12α) ppm. ^13^C NMR (75 MHz, CDCl_3_) δ 155.4 (C-1′), 155.3 (C-1′′), 146.4 (C-8), 146.2 (C-9), 145.4 (C-4), 138.3 (C-4′, C-4′′), 129.4 (C-10a), 128.6 (C-Ar), 128.5(C-Ar), 127.6 (C-Ar), 127.5 (C-Ar), 127.4 (C-Ar), 126.9 (C-6a), 114.6 (C-3), 107.2 (C-7), 105.2 (C-10), 100.9 (-OCH_2_O), 71.6 (C-2), 70.3 (C-1), 56.9 (C-6), 53.6 (C-12), 45.2 (C-3′, C-3′′), 39.7 (C-10b), 28.6 (C-11) ppm. ESI-MS m/z 554 [M + H]^+^.

### 4.3. Biological Assays

#### 4.3.1. Cell Line Cultures

The human colon adenocarcinoma cell lines Colo205 doxorubicin-sensitive (CCL-222, ATCC, Manassas, VA, USA) and the resistant Colo320/MDR-LRP expressing P-gp were purchased from LGC Promochem (Teddington, UK). The MRC-5 human embryonal lung fibroblast cell line (CCL-171, ATCC) was purchased from Sigma-Aldrich (Merck KGaA, Darmstadt, Germany). The human colon adenocarcinoma cells (Colo205 and Colo320) were cultured in an RPMI 1640 medium supplemented with 10% fetal bovine serum (FBS), 2 mM L-glutamine, 1 mM Na-pyruvate, and 100 mM HEPES. The semi-adherent human colon cancer cells were detached with a trypsin-Versene (EDTA) solution for 5 min at 37 °C. The MRC-5 human embryonal lung fibroblast cells were cultured in EMEM containing 4.5 g/L of glucose and supplemented with a non-essential amino acid mixture, a selection of vitamins, and 10% FBS. The cell lines were incubated at 37 °C, 5% CO2, and 95% air atmosphere. 

#### 4.3.2. Antiproliferative Assays

The antiproliferative effect of the compounds was tested in decreasing serial dilutions of compounds (starting with 100 µM, then two-fold serial dilution) on human cell lines (Colo205, Colo320, and MRC-5) in 96-well flat-bottomed microtiter plates. Firstly, the compounds were diluted in 100 µL of the medium, and then 6 × 10^3^ cells (Colo205 and Colo320), in 100 µL of medium, were added to each well, excluding the medium control wells. The MRC-5 adherent cell line was seeded in the EMEM medium for at least 4 h before the assay. The two-fold serial dilutions of the compounds were made in a separate plate (100–0.19 μM) and then transferred to the plates containing the adherent corresponding cell line. Culture plates were incubated at 37 °C for 72 h, and at the end of the incubation period, 20 μL of MTT (thiazolyl blue tetrazolium bromide) solution (from a 5 mg/mL stock solution) was added to each well and incubated for an additional 4 h. Then, 100 μL of sodium dodecyl sulfate (SDS) solution (10% SDS in 0.01 M HCl) was added to each well, and the plates were further incubated at 37 °C overnight in a CO_2_ incubator. Cell growth was determined by measuring the optical density (OD) at 540/630 nm with a Multiscan EX ELISA reader (Thermo Labsystems, Cheshire, WA, USA). The percentage of inhibition of cell growth was determined according to Equation (1) and expressed as IC_50_ values, defined as the concentration that induces 50% growth inhibition. IC_50_ values and the standard deviation (SD) of triplicate experiments were calculated by using GraphPad Prism 5 software for Windows. Doxorubicin (2 mg/mL, Teva Pharmaceuticals, Budapest, Hungary) was used as a positive control and the vehicle DMSO as the negative control.
(1)$$IC_{50}=\left[\frac{OD_{sample}-OD_{medium\;control}}{OD_{cell\;control}-OD_{medium\;control}}\right]\times 100$$

The relative resistance (RR) was calculated as the ratio of the IC_50_ value in the resistant cancer cells and the IC_50_ in the sensitive cancer cell lines. The selectivity indexes (SI) were calculated as the ratio of the IC_50_ values in the non-tumor cells and the IC_50_ in the cancer cell lines. The activity of the compounds toward cancer cells is considered strongly selective if the selectivity index (SI) value is higher than 6, moderately selective if 3 < SI < 6, slightly selective if 1 < SI < 3, and non-selective if SI is lower than 1.

#### 4.3.3. Rhodamine-123 Accumulation Assay

The ability of the tested compound for inhibiting the multidrug P-gp efflux pump was assessed by measuring the accumulation of rhodamine-123 on human Colo205 and Colo320 colon adenocarcinoma cells. The cell number was adjusted to 2 × 10^6^ cells/mL, resuspended in serum-free RPMI-1640 medium, and distributed in 500 µL aliquots. The compounds were tested at different concentrations (0.2, 2, and 20 μM; from 1 mM stock solutions). Verapamil was applied as the positive control (20 μM final), and DMSO was used as the solvent control (at 2 *v*/*v* %). The samples were incubated for 10 min at room temperature. Subsequently, 10 μL (5.2 μM final concentration) of the fluorochrome rhodamine-123 (Sigma, St. Louis, MO, USA) was added to the samples, and the cells were incubated for 20 min at 37 °C, washed twice, and resuspended in 1000 μL of phosphate-buffered serum (PBS) for analysis. The fluorescence of the gated cell population was measured with a Partec CyFlow^®^flow cytometer (Partec, Münster, Germany). The resulting histograms were evaluated regarding mean fluorescence intensity (FL-1), standard deviation, both forward scatter count (FSC) and side scatter count (SSC) parameters, and the peak channel of 20 000 individual cells belonging to the total and the gated populations (Appendix A). The percentage of the mean fluorescence intensity (FL-1) was calculated for the treated MDR cells compared to the untreated cells. The fluorescence activity ratio (FAR) was calculated based on Equation (2), which relates the measured fluorescence values:(2)$$FAR=\frac{\;Colo320_{treated}/Colo320_{control}\;}{Colo205_{treated}/Colo205_{control}}$$

#### 4.3.4. Drug Combination Assay

Drug combination interactions between the compounds and the chemotherapeutic drug doxorubicin were assessed by the checkboard microplate assay on the resistant Colo320, expressing the P-gp transporter. Doxorubicin was serially diluted in the horizontal direction in 100 µL, starting with 8.6 µM, and the compounds were subsequently diluted in the vertical direction in 50 µL. The starting concentration was determined based on the IC50 values. The cells were resuspended in a culture medium and distributed into each well in 50 µL, containing 6 × 10^3^ cells, to a final volume of 200 µL per well. The plates were incubated for 72 h at 37 °C in a CO_2_ incubator, and the cell growth was determined by the MTT staining method, as described previously. Drug interactions were evaluated using CompuSyn software to plot 4 or 5 data points for each ratio (ref software). The results were expressed in terms of combination index (CI) values of 50% growth inhibition, in which a CI value close to 1 indicates additivity, while a CI < 1 is defined as synergy, and a CI > 1 as antagonism. 

#### 4.3.5. ATPase Activity Assay

The P-glycoprotein ATPase assay was assessed using the P-gp-Glo^TM^ Assay System (Promega, WI, USA). The assay was performed according to the manufacturer’s instructions [29]. Briefly, 20 µL of recombinant human P-gp membranes (1.25 mg/mL), expressing high levels of human P-gp, were incubated in 20 µL of P-gp-Glo^TM^ assay buffer for 5 min at 37 °C. Compounds were tested at 25 µM, sodium orthovanadate (Na_3_VO_4_, 0.25 mM) was applied as an inhibitor control, verapamil was used as substrate control (0.5 mM), and DMSO at 2% was applied as solvent control. The reaction was initiated by adding 10 µL of 25 mM MgATP, and incubated at 37 °C for 40 min. The reaction was stopped after adding 50 µL of ATP Detection Reagent, and the samples and controls were incubated at room temperature for 20 min. The emitted luciferase-generated luminescent signal was measured in a CLARIOstar Plus plate reader (BMG Labtech, Ortenberg, Germany) at 580 nm. The relative ATPase activity was calculated based on the ratio between the luminescence measured of the P-gp ATPase activity of each compound and the basal P-gp ATPase activity according to Equation (3): (3)$$\mathrm{Relative}\;\mathrm{ATPase}\;\mathrm{activity}=\frac{\;Lum_{treated}-Lum_{untreated}\;}{Lum_{Na3VO4}/Lum_{untreated}}$$

The effect of the tested compounds was evaluated according to the instructions of the manufacturer. 

### 4.4. Statistical Analyses

The statistical data were analyzed using GraphPad Prism (La Jolla, CA, USA; version 5) software for Microsoft Windows 10. For the comparison of two groups, the unpaired Student’s *t*-test was used. Statistical significance was set as * *p* < 0.05, ** *p* < 0.01.

## 5. Conclusions

Aiming at finding new P-gp-mediated MDR reversing compounds, this study focused on the generation of a new set of Amaryllidaceae-type alkaloids by the derivatization of lycorine (**1**), one of the major alkaloids isolated from *P. maritimum*. In this way, by reaction of the free hydroxyl groups of this compound, thirty-one new mono- and di-carbamates (**2**–**32**) were prepared. They were evaluated as MDR reversers in human colon adenocarcinoma cells, by functional and chemosensitivity assays. Most of the derivatives showed an increase in the P-gp inhibitory activity, when compared with the parental compound, in non-cytotoxic concentrations. The strongest inhibitors were the di-carbamates bearing phenethyl (**23**, **25**) or benzyl moieties (**5**, **9**, **20**, **22**, **27**, **32**), being more active than verapamil. Moreover, some derivatives (**5**, **9**, and **25**) showed significant selective antiproliferative activity in P-gp overexpressing cells, having a collateral sensitivity effect, thus a dual role in reversing P-gp-mediated MDR. In the chemosensitization assay, all the tested derivatives interacted synergistically with doxorubicin, corroborating the results obtained in the transport assay. In the ATPase activity assay, selected compounds behaved as inhibitors.

## Data Availability

The data that support the findings of this study are available from the corresponding author upon request.

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
