# Peer review of "Lycorine Carbamate Derivatives for Reversing P-glycoprotein-Mediated Multidrug Resistance in Human Colon Adenocarcinoma Cells"

_ijms, 2023, doi:10.3390/ijms24032061_

Round 1
Reviewer 1 Report
The paper is dedicated to searching for anticancer compounds on the basis of the modification of alkaloid lycorine. 32 compounds were synthesized and evaluated by some methods. English is very good. The paper is read well
However, some moments need to be specified more clearly.
1. Explain why compounds with RR were not exceeding 0.5 show collateral sensitivity (CS). Which is a connection?
2. Inhibition of P-glycoprotein efflux activity. The connection between FAR and the conclusion about the Inhibition of P-glycoprotein efflux activity is unclear.
3. Line 341. Log P or log S. Specify
4. Chemosensitivity assays. Add 2-3 sentences that explain the sense of this test and explain criteria - strong synergism,synergism, etc
Author Response
Reviewer #1
The paper is dedicated to searching for anticancer compounds on the basis of the modification of alkaloid lycorine. 32 compounds were synthesized and evaluated by some methods. English is very good. The paper is read well. However, some moments need to be specified more clearly.
- Explain why compounds with RR were not exceeding 0.5 show collateral sensitivity (CS). Which is a connection?
Answer: We are following the collateral sensitivity concept as described by Hall et. al. The relative resistance (RR) is determined by dividing the IC50 against the resistant subline by the IC50 in the corresponding parental line. A CS agent shows greater toxicity against the MDR line than against the parental line and the RR is <1. However, at least a two-fold effect seems to be required to be significant. Therefore, it is considered that a compound exhibits collateral sensitivity effect if RR ≤ 0.5.
Hall, M.D.; Handley, M.D.; Gottesman, M.M. Is Resistance Useless? Multidrug Resistance and Collateral Sensitivity. Trends Pharmacol. Sci. 2009, 30, 546–556, doi:10.1016/j.tips.2009.07.003.
- Inhibition of P-glycoprotein efflux activity. The connection between FAR and the conclusion about the Inhibition of P-glycoprotein efflux activity is unclear.
Answer: We have completed the sentence for clarifying: “Compounds with FAR values higher than 1 are able to decrease rhodamine-123 efflux in resistant cells and are considered P-gp inhibitors”
In fact, fluorescence activity ratio (FAR) values are a measure of the cytosolic accumulation of rhodamine-123 (substrate of P-gp) between resistant and sensitive cells. As mentioned in the manuscript, FAR values are calculated by determining the quotient between the intracellular accumulation of rhodamine 123 in resistant and sensitive cancer cells. Thus, a compound with FAR values higher than 1 means that it is able to inhibit the efflux of rhodamine 123 by P-gp.
- Line 341. Log P or log S. Specify
Answer: As defined at pkCSM web server (https://biosig.lab.uq.edu.au/pkcsm/prediction), Log S is the solubility of a compound in water at 25 ºC. This definition was included in the manuscript.
- Chemosensitivity assays. Add 2-3 sentences that explain the sense of this test and explain criteria - strong synergism,synergism, etc
Answer: As suggested by the reviewer some sentences were included in the manuscript. These data are detailed in Figure 1

Reviewer 2 Report
The study aims to evaluate effect of newly prepared 32 lycorine carbamate derivatives on multidrug resistance reversal and synergy with doxorubicine. The study is well designed, results presented in clear way and supporing the conclusions. I did not detect major flaws or problems in the study. Whether this type of study (medicinal chemistry) fits the scope of IJMS I am, however, not sure.
Author Response
Reviewer # 2
The study aims to evaluate effect of newly prepared 32 lycorine carbamate derivatives on multidrug resistance reversal and synergy with doxorubicine. The study is well designed, results presented in clear way and supporing the conclusions. I did not detect major flaws or problems in the study. Whether this type of study (medicinal chemistry) fits the scope of IJMS I am, however, not sure.
Answer: We thank reviewer #2 for the positive comments